# LRP5 promotes adipose progenitor cell fitness and adipocyte insulin sensitivity

Nellie Y. Loh [1], Senthil K. Vasan[1], Daniel B. Rosoff[1,2,3], Emile Roberts [4], Andrea D. van Dam[1], Manu Verma[1], Daniel Phillips [1], Agata Wesolowska-Andersen[5], Matt J. Neville [1,6], Raymond Noordam [7], David W. Ray[1,6], Jonathan H. Tobias [2,8], Celia L. Gregson [2,8], Fredrik Karpe [1,6] & Constantinos Christodoulides [1,6] ✉

## Abstract

**Background** WNT signaling plays a key role in postnatal bone formation. Individuals with gain-of-function mutations in the WNT co-receptor LRP5 exhibit increased lower-body fat mass and potentially enhanced glucose metabolism, alongside high bone mass. However, the mechanisms by which LRP5 regulates fat distribution and its effects on systemic metabolism remain unclear. This study aims to explore the role of LRP5 in adipose tissue biology and its impact on metabolism.

**Methods** Metabolic assessments and imaging were conducted on individuals with gain- and loss-of-function *LRP5* mutations, along with age- and BMI-matched controls. Mendelian randomization analyses were used to investigate the relationship between bone, fat distribution, and systemic metabolism. Functional studies and RNA sequencing were performed on abdominal and gluteal adipose cells with LRP5 knockdown.

**Results** Here we show that LRP5 promotes lower-body fat distribution and enhances systemic and adipocyte insulin sensitivity through cell-autonomous mechanisms, independent of its bone-related functions. LRP5 supports adipose progenitor cell function by activating WNT/β-catenin signaling and preserving valosin-containing protein (VCP)-mediated proteostasis. *LRP5* expression in adipose progenitors declines with age, but gain-of-function *LRP5* variants protect against age-related fat loss in the lower body.

**Conclusions** Our findings underscore the critical role of LRP5 in regulating lower-body fat distribution and insulin sensitivity, independent of its effects on bone. Pharmacological activation of LRP5 in adipose tissue may offer a promising strategy to prevent age-related fat redistribution and metabolic disorders.

## Plain language summary

This study investigated how a protein called LRP5 affects where fat is stored in the body and how it influences metabolism (how the body uses nutrients for energy). Some people with specific changes (mutations) in the *LRP5* gene tend to store more fat in their lower body, which is protective against conditions such as diabetes and heart disease, while fat in the upper body can increase health risks. To understand this, we studied individuals with different *LRP5* mutations and conducted various tests and cell studies. We found that LRP5 plays a key role in directing fat to the lower body and helps fat cells respond better to insulin, independent of its role in bone health. These findings suggest that targeting LRP5 could help prevent unhealthy fat storage with aging and improve metabolic health.

The WNT family of secreted glycoproteins play essential roles during adult tissue homeostasis by engaging multiple intracellular signaling cascades[1]. In the canonical pathway, WNT binding to low-density lipoprotein (LDL)-related protein (LRP)5 and LRP6 co-receptors regulates WNT target gene expression by inhibiting glycogen synthase kinase 3 (GSK3)-dependent degradation of the transcriptional co-regulator β-catenin[1]. GSK3 is a promiscuous kinase and WNT signaling has also been shown to protect other GSK3 target proteins from degradation, including c-MYC and cyclin-D1. This transcription-independent signaling pathway has been termed WNT-dependent stabilization of proteins (WNT/STOP)[2]. WNT signaling is indispensable for normal postnatal bone accrual. Rare, homozygous, loss-of-function (LoF) *LRP5* mutations in humans cause profound

[1]Radcliffe Department of Medicine, Oxford Centre for Diabetes, Endocrinology and Metabolism, University of Oxford, Oxford, UK. [2]MRC Integrative Epidemiology Unit (IEU), Population Health Sciences, Bristol Medical School, University of Bristol, Bristol, UK. [3]National Institute on Alcohol Abuse and Alcoholism, National Institutes of Health, Bethesda, MD, USA. [4]Sir William Dunn School of Pathology, University of Oxford, Oxford, UK. [5]Nuffield Department of Medicine, Wellcome Centre for Human Genetics, University of Oxford, Oxford, UK. [6]NIHR Oxford Biomedical Research Centre, OUH Foundation Trust, Oxford, UK. [7]Department of Internal Medicine, Section of Gerontology and Geriatrics, Leiden University Medical Center, Leiden, The Netherlands. [8]Musculoskeletal Research Unit, Translational Health Sciences, Bristol Medical School, Southmead Hospital, University of Bristol, Bristol, UK. ✉e-mail: costas.christodoulides@ocdem.ox.ac.uk

osteoporosis[3], a phenotype recapitulated in mice with germline or osteocyte-specific *Lrp5* deletion[4]. Conversely, rare heterozygous gain-of-function (GoF) *LRP5* mutations lead to inherited syndromes of high bone mass (HBM)[5–7].

In addition to the well-established contribution of LRP5 in skeletal homeostasis, human and animal studies have highlighted potential roles for this receptor in white adipose tissue (WAT) biology and systemic metabolism. LRP5 knockdown (KD) in 3T3-L1 preadipocytes blocked adipogenesis[8] whilst in vivo, *Lrp5* null mice displayed impaired glucose tolerance and high-fat diet (HFD)-induced hypercholesterolemia[9]. In humans, WAT *LRP5* expression was diminished in insulin-resistant subjects[10] and *LRP5*-KD in adipose progenitors (APs) led to dose- and depot-dependent effects on in vitro adipogenesis[11]. Furthermore, subjects with rare LoF *LRP5* mutations and osteoporosis had a higher prevalence of type 2 diabetes[12].

We previously showed that alongside HBM, GoF *LRP5* mutations were associated with lower-body fat distribution and potentially improved systemic insulin sensitivity[11]. However, a subsequent study found no causal effects of GoF *LRP5* mutations on glucose homeostasis[13]. Hence, the role of LRP5 in systemic metabolism remains controversial. Additionally, the cellular and molecular mechanisms driving the effects of LRP5 on fat distribution and potential impacts on whole-body metabolism remain unmapped. To address these questions, we conducted in vivo studies involving GoF and LoF *LRP5* variant carriers, Mendelian randomization (MR) analyses, functional assays using human adipocytes and APs with induced *LRP5*-KD, and genome-wide transcriptomic analyses in *LRP5*-KD APs. We demonstrate that LRP5 promotes lower-body fat distribution by supporting AP function and proteostasis and improves systemic and adipocyte insulin sensitivity. These effects are independent of LRP5's role in bone. Our findings suggest that activating LRP5 in WAT could be a promising strategy to prevent age-related lower-body fat loss and associated metabolic disorders.

## Methods
### Study participants
Study participants were recruited from the Oxford Biobank (OBB)[14] and from the UK-wide HBM cohort[11,15]. Fasting blood sampling, anthropometric and DXA measurements were undertaken. A sub-group also underwent oral glucose tolerance tests (OGTTs). WAT biopsies were obtained by needle biopsy from the periumbilical and buttock areas and cell fractionations were performed by collagenase treatment and centrifugation[11]. All studies were approved by the Oxfordshire Clinical Research Ethics Committee. All volunteers gave written informed consent.

### Cell lines
Immortalized APs were generated in-house by transgenesis of primary APs from a male donor with human telomerase reverse transcriptase and HPV-E7 oncoprotein. The donor, an OBB participant, provided written informed consent. Ethical approval for the fat biopsies in OBB participants was obtained from the Oxfordshire Clinical Research Ethics Committee. De-differentiated fat (DFAT) cells were derived by selection and de-differentiation of lipid-laden, in vitro differentiated immortalized human APs[16,17]. Immortalized APs stably expressing scrambled (shCON) and LRP5 (shLRP5) shRNAs were previously published[11].

### Doxycycline-inducible cell lines
Oligonucleotides for shLRP5 (top: 5′CCGGGACGCAGTACAGCGATTATATCTCGAGATATAATCGCTGTACTGCGTCTTTTT; bottom: 5′AATTAAAAAGACGCAGTACAGCGATTATATCTCGAGATATAATCGCTGTACTGCGTC), shVCP#1 (top: 5′ CCGGGAATAGAGTTGTTCGGAATAACTCGAGTTATTCCGAACAACTCTATTCTTTTT; bottom: 5′AATTAAAAAGAATAGAGTTGTTCGGAATAACTCGAGTTATTCCGAACAACTCTATTC), shVCP#4 (top: 5′ CCGGAGATCCGTCGAGATCACTTTGCTCGAGCAAAGTGATCTCGACGGATCTTTTTT;

bottom: 5′AATTAAAAAAGATCCGTCGAGATCACTTTGCTCGAGCAAAGTGATCTCGACGGATCT), and shCON (top: 5′CCGGCAACAAGATGAAGAGCACCAACTCGAGTTGGTGCTCTTCATCTTGTTGTTTTT; bottom: 5′AATTAAAAACAACAAGATGAAGAGCACCAACTCGAGTTGGTGCTCTTCATCTTGTTG) were annealed and cloned into the tet-pLKO-puro doxycycline-inducible expression lentiviral vector (gift from Dmitri Wiederschain, Addgene #21915)[18]. DFAT cells stably expressing tet-pLKO-puro-shLRP5 (tet-shLRP5), tet-pLKO-puro-shVCP#1 (tet-shVCP#1), tet-pLKO-puro-shVCP#4 (tet-shVCP#4) and tet-pLKO-puro-shCON (tet-shCON) were generated by lentiviral transduction and selection in 2 μg/ml puromycin, and maintained and plated under tetracycline-free conditions[16] (Supplementary Methods).

For shRNA-induction, plated APs were treated with doxycycline (or vehicle) for ~48–96 h, or differentiated 2 days post-plating in the presence of 0.05 μg/ml doxycycline (or vehicle) throughout[11]. For shRNA-induction in in vitro differentiated adipocytes, cells were differentiated for 13 days, then incubated in hormone-free basal media containing 0.05 μg/ml doxycycline (or vehicle) for 48 h[16]. Intracellular lipids were quantified using AdipoRed (Lonza) and a PHERAstar *FS* microplate reader (BMG Labtech).

The SLC2A3 full-length open-reading-frame (gift from William Hahn, David Root and Jesse Boehm, Addgene #81787)[19] was cloned into the pLenti-CMVtight-Hygro-DEST (pLENTI) lentivector (gift from Guillermo de Cárcer, Addgene #136339). Tet-shCON and tet-shLRP5 cells co-expressing the pLENTI-SLC2A3 (or empty) vector were generated by lentiviral transduction and selection in 20 μg/ml hygromycin B. shRNA and SLC2A3 expression were induced with 0.02 μg/ml doxycycline.

### Supplementation and inhibition experiments
For supplementation experiments, 5 mM sodium pyruvate (Merck), Wnt surrogate Fc-fusion recombinant protein (Thermo Fisher Scientific, PHG0401), or vehicle were added to adipogenic media, with or without doxycycline. For GSK3 and VCP inhibition experiments, indicated concentrations of the GSK3 inhibitor CHIR99021 (Abcam) (with or without doxycycline), VCP inhibitors DBeQ (APExBio, cat#A8629) or NMS-873 (Cambridge Bioscience, cat#CAY17674), or vehicle (DMSO) were added to the growth or adipogenic media. To assess CHIR99021 effects in G2/M-arrested cells, ~50 K cells seeded in 6-well plates were treated the next day with 0.1 μg/ml doxycycline (or vehicle) for ~24 h, then a further 24 h with media containing 100 ng/ml Nocodazole (or DMSO) with or without 0.1 μg/ml doxycycline, followed by a 1 h-treatment with 0.5 μM CHIR99021 (or vehicle).

### Proliferation assays
AP proliferation was assessed in 96-well plates using the CyQUANT® Direct Proliferation Assay (Thermo Fisher Scientific) and a PHERAstar *FS* microplate reader[11]. Doubling time was calculated using the formula: $T_d = (t_2 - t_1) \times [\log\ (2) \div \log\ (q_2 \div q_1)]$, where $t$ = time (days), $q$ = fluorescence intensity (surrogate for cell number).

### Clonogenic potential
Single cells were flow-sorted into 96-well plates (1 cell/well) and cultured for 15 days in conditioned media containing doxycycline or vehicle. Cell density was assessed using the CyQUANT® Direct assay and cell number was estimated from a standard curve.

### Apoptosis assays
Cells were cultured for 3 days in growth media containing doxycycline or vehicle, then a further 24 h in serum-free media containing doxycycline or vehicle. Apoptosis was assayed using Caspase-Glo 3/7 assay (Promega) and a Veritas Microplate Luminometer (Turner Biosystems). Results were normalized to cell number.

### Glucose uptake assays
In vitro differentiated cells were incubated in either fresh hormone-free basal medium (to measure basal uptake) or basal medium containing 25 nM

insulin for 30 min at 37 °C, 5%CO$_2$. Cells were then washed twice in HEPES-buffered saline (HBS; 140 mM NaCl, 20 mM HEPES, 5 mM KCl, 2.5 mM MgSO$_4$, 1 mM CaCl, pH7.4), incubated for 10 min at room temperature in uptake buffer (10 μM 2-deoxy-D-glucose and 0.024MBq/ml 2-[$^3$H]-deoxy-D-glucose in HBS), washed twice in ice-cold 0.9% NaCl, and lysed in 1.2 ml 50 mM NaOH. Radioactivity was measured using 1 ml lysate mixed with 4 ml liquid scintillant (Perkin Elmer) in a Beckman LS6500 Multipurpose Scintillation Counter (Beckman). Results were corrected for nonspecific diffusion (cells incubated in uptake buffer containing 10 μM Cytochalasin B), and normalized to protein concentration[16]. For APs, only basal glucose uptake was measured.

### TOPflash reporter assay
Tet-shCON and tet-shLRP5 cells co-expressing the 7TFC TOPflash reporter vector (gift from Roel Nusse, Addgene #24307)[20] were treated with doxycycline or vehicle for 96 h. TOPflash reporter activity was measured using the Luciferase Assay System (Promega) on a Veritas Luminometer, and normalized to mCherry fluorescence. To assess the effects of VCP inhibitor treatment on WNT/β-catenin signaling, APs transduced with 7TFP[11] were seeded in 24-well plates and treated with indicated concentrations of DBeQ, NMS-873, or vehicle in serum-free media for 24 h prior to measuring TOPflash reporter activity. Results were normalized to protein concentration.

### RNA sequencing
RNA sequencing of vehicle and doxycycline-treated tet-shCON and tet-shLRP5 APs (three independent experiments) was performed at the Oxford Genomics Centre (WTCHG, Oxford, UK) (Supplementary Information). Differentially expressed genes (DEGs) were identified using the DESeq2 v1.34.0 R package[21]. Gene-set enrichment analysis (GSEA) was performed on DEGs with padj < 0.05 in Metascape[22]. Transcription factor binding-site motif analyses of DEGs were performed using iRegulon in Cytoscape[23].

### Quantitative real-time PCR and western blotting
Taqman assays and antibodies used are listed in Supplementary Table 1.

### Mendelian randomization
We performed two-sample MR to investigate the relationship between heel-estimated bone mineral density (eBMD) and anthropometric and metabolic traits. Exposure instruments were extracted from heel eBMD GWAS summary statistics from the UK Biobank (UKB)[24] using extract_instruments() in TwoSampleMR ($p < 5e-8$, LD $r^2 < 0.001$, genetic distance = 10 Mb, 1000 Genomes Phase 3 European population reference panel). A minor allele frequency (MAF) threshold of 0.01 was used for instrumental variable (IV) selection. As outcome data we used the largest publicly available GWAS summary statistics for anthropometric traits (body mass index (BMI) and BMI-adjusted waist-to-hip ratio (WHRadjBMI)[25], BMI-adjusted waist and hip circumference (WCadjBMI, HIPadjBMI)[26]), MRI-derived visceral, abdominal subcutaneous, and gluteofemoral WAT volumes, adjusted for BMI and height (vatadjbmi3, asatadjbmi3, gfatadjbmi3)[27], and glycemic (BMI-adjusted fasting glucose and insulin[28]) and lipid[29] traits (Supplementary Table 2). We performed inverse variance weighted (IVW)[30] MR, with MR-Egger[31] and weighted-median[32] as sensitivity analyses, using the TwoSampleMR package (v0.5.7) in R (v4.3.1)[33]. Results with a $p < 0.05$ in the primary IVW MR analysis and in at least one of the two sensitivity analyses were considered evidence for causal association. As additional sensitivity analysis, we repeated the MRs using IVs with MAF ≥ 0.05.

### Statistical analyses
Statistical analyses for the human studies were carried out using SPSS. Statistical analyses for in vitro studies and graph generation were done in GraphPad, while MR and RNA-seq analyses were carried out in R. Statistical tests used are stated within figure legends and table footnotes, and detailed in Supplementary Information.

### Reporting summary
Further information on research design is available in the Nature Portfolio Reporting Summary linked to this article.

## Results

### LRP5 and systemic metabolism
To gain further insights into the role of LRP5 in systemic metabolism, we re-evaluated the glucose and lipid profiles of six individuals with HBM due to rare heterozygous GoF *LRP5* mutations (LRP5$_{A242T}$ and LRP5$_{N198S}$) (Table 1 and Supplementary Tables 3, 4). Each subject was age- and BMI-matched to ten healthy volunteers. Compared to controls, GoF *LRP5* mutation carriers exhibited lower fasting glucose, fasting insulin, Homeostatic Model Assessment for Insulin Resistance (HOMA-IR), HOMA of β-cell function (HOMA-B), and adipose tissue insulin resistance (Adipo-IR). We also examined the metabolic profile of 23 homozygous carriers of rs4988321, a low-frequency, missense *LRP5* variant (LRP5$_{V667M}$) presumed to be LoF since it was shown to be associated with lower heel eBMD in a large GWAS[24] and to account for 100% of the posterior probability of the association at this signal[34]. Each subject was again matched to ten healthy controls. Compared to controls, LoF *LRP5* cases displayed nominally higher fasting insulin, HOMA-IR, Adipo-IR, and triglycerides (Table 1, Supplementary Table 5 and Supplementary Data 1). Finally, we conducted OGTTs in the GoF *LRP5* cases and ten independent controls (Supplementary Table 6). In this smaller cohort, glucose and insulin levels during the OGTT were similar between cases and controls, but GoF *LRP5* mutation carriers exhibited reduced post-OGTT non-esterified fatty acid (NEFA) levels (Fig. 1 and Supplementary Table 6). In conclusion, LRP5 positively regulates glucose and lipid metabolism. Additionally, the increased insulin sensitivity associated with GoF *LRP5* mutations might be partly driven via enhanced insulin action in WAT.

### LRP5 and fat distribution
Compared to controls, GoF *LRP5* variant carriers had markedly higher total BMD. Moreover, despite no differences in total fat mass, they exhibited higher leg fat mass and a lower android-to-leg fat mass ratio (Table 1). Conversely, LoF *LRP5* cases displayed lower total BMD and leg fat mass than controls. Consistent with these findings, subcutaneous WAT *LRP5* expression negatively correlated with the android-to-leg fat ratio in both abdominal and gluteal WAT in women, and in gluteal WAT in men (Supplementary Fig. 1a–d). To gain mechanistic insights of how LRP5 influences regional adiposity, we analyzed *LRP5* expression in fractionated WAT from another cohort of 43 females with available DXA data (Table 2 and Supplementary Table 7). In age- and percent fat mass-adjusted partial correlations, both abdominal and gluteal AP *LRP5* expression positively correlated with lower-body fat mass. No robust correlations between mature adipocyte *LRP5* expression and regional adiposity were identified. Finally, *LRP5* mRNA abundance was higher in the stromovascular (SVF) than the adipocyte fraction of both subcutaneous abdominal and gluteal WAT, with the highest expression within the AP fraction (Supplementary Fig. 1e–g). These findings collectively underscore that LRP5 promotes a lower-body fat distribution and point to APs as the likely effector cells.

### Mendelian randomization of BMD and metabolic and anthropometric traits
The primary phenotype associated with the described functional *LRP5* variants was a change in bone mass (Table 1). As the skeleton has been reported to be a site of high glucose and NEFA uptake[35,36] and to influence systemic metabolism through the secretion of hormones such as osteocalcin[35], we investigated the impact of BMD on systemic metabolism and fat distribution using two-sample MR. As IVs, we utilized all the independent single nucleotide variations (SNVs) located throughout the genome that were significantly associated with heel eBMD in a UKB GWAS[24]. Subsequently, we extracted the effect estimates for glycemic[28], lipid[29] and adiposity[25–27] traits for each SNV from the largest publicly available GWAS. In univariate IVW MR analyses, higher heel eBMD had a

**Table 1 | Comparison of plasma biochemistry and body composition (DXA) of individuals with *LRP5* gain-of-function (GoF) (A242T, N198S) and *LRP5* loss-of-function (LoF) (V667M) variants and age, sex and BMI-matched controls**

| | *LRP5* GoF | | *LRP5* LoF | |
|---|---|---|---|---|
| | Mean difference (95% CI) | *P* value | Mean difference (95% CI) | *P* value |
| Sex, case (controls) | 4F, 2M (40F, 20M) | | 15F, 8M (150F, 79M) | |
| Age (years)[a] | 2.5 ± 8.3 | | 0.7 ± 2.8 | |
| BMI (kg/m$^2$)[a] | −0.08 ± 0.20 | | −0.01 ± 0.04 | |
| Z-height | 0.89 (0.09, 1.69) | 0.029 | −0.45 (−0.86, −0.04) | 0.030 |
| Z-weight | 0.88 (0.08, 1.68) | 0.032 | −0.45 (−0.86, −0.04) | 0.030 |
| DXA measurements[b] | | | | |
| Z-Fat android | −0.49 (−1.31, 0.32) | 0.23 | 0.05 (−0.48, 0.59) | 0.84 |
| Z-Fat gynoid | 0.23 (−0.60, 1.06) | 0.23 | −0.51 (−1.03, 0.02) | 0.057 |
| Z-Fat visceral | −0.30 (−1.13, 0.52) | 0.47 | 0.09 (−0.44, 0.62) | 0.73 |
| Z-Fat legs | 0.89 (0.09, 1.69) | 0.029 | −0.59 (−1.11, −0.06) | 0.028 |
| Z-Total fat mass | 0.45 (−0.37, 1.27) | 0.28 | −0.10 (−0.63, 0.43) | 0.71 |
| Z-Total fat percentage | −0.23 (−1.06, 0.59) | 0.57 | −0.16 (−0.69, 0.37) | 0.55 |
| Z-Android/leg fat ratio | −1.05 (−1.40, −0.70) | 3.29 × 10$^{-6}$ [c] | 0.42 (−0.10, 0.95) | 0.11 |
| Z-Total lean mass | 0.34 (−0.48, 1.16) | 0.41 | −0.00 (−0.53, 0.53) | 0.99 |
| Z-Lean mass legs | 0.12 (−0.71, 0.95) | 0.77 | 0.02 (−0.51, 0.56) | 0.93 |
| Z-Total BMD | 3.11 (2.99, 3.24) | 2.18 × 10$^{-25}$ [c] | −0.73 (−1.24, −0.21) | 0.006 |
| Biochemistry | | | | |
| Z-glucose (fasting) | −0.96 (−1.75, −0.16) | 0.019 | −0.11 (−0.53, 0.30) | 0.58 |
| Z-insulin (fasting) | −0.92 (−1.71, −0.12) | 0.025 | 0.40 (−0.01, 0.81) | 0.06 |
| Z-HOMA-IR | −0.92 (−1.72, −0.13) | 0.024 | 0.39 (−0.02, 0.79) | 0.06 |
| Z-HOMA-B | −0.63 (−1.05, −0.21) | 0.006[c] | 0.34 (−0.07, 0.75) | 0.11 |
| Z-NEFA | −0.24 (−1.07, 0.59) | 0.57 | 0.24 (−0.17, 0.65) | 0.25 |
| Z-Adipo-IR | −0.81 (−1.25, −0.38) | 0.001[c] | 0.36 (−0.05, 0.77) | 0.09 |
| Z-triglycerides | −0.58 (−1.39, 0.24) | 0.16 | 0.38 (−0.03, 0.79) | 0.07 |

Absolute values are shown in Supplementary Tables 3 and 5. Data represent mean difference (case-control) in outcome (Z-transformed) for each case within the cluster of ten age, sex and BMI-matched controls.

*Adipo-IR* adipose tissue insulin resistance, *BMD* bone mineral density, *BMI* body mass index, *CI* confidence interval, *DXA* dual-energy X-ray absorptiometry, *HOMA-IR* Homeostatic Model Assessment for Insulin Resistance, *HOMA-B* Homeostatic Model Assessment for Insulin Secretion, *NEFA* non-esterified fatty acids.

[a]Represent mean and SD of the case from its controls within each cluster.

[b]For LoF, DXA measurements were calculated for 14 cases (10F, 4M) and their matched 139 controls on whom data were available.

[c]*P* value obtained by two-sample *t*-test ([c]with Welch's correction for unequal variance).

negative impact on HIPadjBMI ($\beta \pm SE = -0.048 \pm 0.018$, $p = 0.008$) and MRI-determined GFAT volume adjusted for BMI and height ($\beta \pm SE = -0.102 \pm 0.020$, $p = 6.41E-7$), and a positive impact on BMI-adjusted fasting insulin ($\beta \pm SE = 0.014 \pm 0.006$, $p = 0.01$) (Table 3, Supplementary Table 8). Sensitivity analyses, including MR analyses using heel eBMD IVs with MAF ≥ 0.05, produced consistent results and found no evidence of unbalanced pleiotropy. However, directionally consistent effect estimates were observed only for the associations between heel eBMD and adiposity traits. Furthermore, at least one out of the two sensitivity analyses was significant for gfatadjbmi3 and fasting insulin. These data suggest that the metabolic and adipose phenotypes of *LRP5* variant carriers are independent of changes in bone mass/biology.

## LRP5 cell autonomously regulates adipocyte insulin sensitivity

To determine if LRP5 can directly regulate insulin action in adipocytes, we induced *LRP5*-KD using a Tet-On system in immortalized DFAT cells[16,17]. LRP5 depletion in differentiated abdominal and gluteal adipocytes following 48-h doxycycline treatment was highly efficient (>90% at the protein level) (Fig. 2a–c and Supplementary Figs. 2a, 3a), and did not lead to changes in lipid accumulation or alterations in adipogenic, adipocyte marker, and insulin pathway gene expression (Fig. 2d, e and Supplementary Fig. 2), except for a notable reduction in adiponectin (*ADIPOQ*) mRNA abundance (Fig. 2d and Supplementary Fig. 2d). In functional assays, LRP5 depletion

was associated with reduced basal glucose uptake and decreased *SLC2A1* and/or *SLC2A3*, encoding the glucose transporters GLUT1 and GLUT3, in both abdominal and gluteal adipocytes. Additionally, it impaired insulin-stimulated glucose uptake and AKT phosphorylation selectively in abdominal adipocytes (Fig. 2f–h and Supplementary Figs. 2k, l, 3b). In complementary experiments, *LRP5* expression exhibited a strong positive correlation with *ADIPOQ* expression in primary mature adipocytes (Fig. 2i, j). We conclude that LRP5 directly regulates adipocyte glucose uptake, insulin sensitivity and *ADIPOQ* expression.

## Transcriptome-wide profiling of LRP5 knockdown APs

To determine the genes and biological processes regulated by LRP5 in APs, we undertook RNA sequencing in DFAT cells with induced *LRP5*-KD (Fig. 3a, b and Supplementary Fig. 4). *LRP5*-KD resulted in altered expression of 871 and 955 genes in abdominal and gluteal APs, respectively, with 584 (>60%) of DEGs being shared. The majority of DEGs, including all the top 30, were downregulated in both abdominal and gluteal cells (Fig. 3c). Gene ontology (GO) enrichment analysis revealed that the downregulated genes in *LRP5*-KD APs were enriched for pathways and processes involved in the cell cycle and carbohydrate metabolism (Fig. 3d, e). The latter category included *SLC2A3*, encoding the high-affinity glucose transporter GLUT3, as well as genes involved in multiple steps of glycolysis and de novo lipogenesis (Fig. 3f and Supplementary Fig. 5a, b). *LRP5*-KD was also associated with

reduced expression of genes involved in ossification and WNT signaling. The upregulated genes were enriched for pathways and processes related to actin cytoskeleton and extracellular matrix organization. One of the most induced genes was *DKK1*, encoding a potent, extracellular LRP5/6 antagonist (Fig. 3f and Supplementary Fig. 5c). Additionally, transcription factor (TF)-binding site motif analysis of the promoters of DEGs in *LRP5*-KD APs (Fig. 3d, e) indicated that the promoters of downregulated genes were enriched for binding sites of multiple TFs involved in cell cycle

regulation, including several E2F family members, TFDP1, which co-operatively regulates cell cycle genes with E2Fs, and FOXM1. In contrast, the promoters of upregulated genes were enriched for motifs bound by TFs involved in the stress response (e.g., FOS, JUN, ETS), as well as motifs for SRF and multiple TEAD TF family members, which co-operatively restrain adipogenesis by transducing cytoskeletal tension-generated mechan-osensitive signaling[37,38]. Overall, these data underscore the crucial role of LRP5 in AP biology.

### LRP5 depletion compromises AP fitness

Consistent with the GO enrichment analyses, induced *LRP5*-KD impaired proliferation in both abdominal and gluteal DFAT APs (Fig. 4a). However, LRP5 depletion only resulted in mild mitotic defects (Supplementary Fig. 6) and was not associated with abnormal chromosome segregation (our unpublished data), which have been linked to impaired WNT/STOP signaling[39]. *LRP5*-KD also led to increased apoptosis and impaired adipo-genesis in both abdominal and gluteal APs (Fig. 4b, c). Time-course studies revealed that LRP5 exerts its pro-adipogenic effects mainly during early adipogenesis (Supplementary Fig. 7a). Next, we investigated the impact of *LRP5*-KD on the clonogenic potential of APs, i.e., their ability to 'infinitely' produce progeny, a measure stem cell status. *LRP5*-KD led to a dramatic reduction in the ratio of colonies formed to the number of cells seeded (Fig. 4d). *LRP5*-KD DFAT cells also formed much smaller colonies than their control counterparts did. Finally, we explored which signaling path-way(s) mediate the effects of LRP5 on AP biology. *LRP5*-KD led to diminished active β-catenin protein levels and reduced TOPflash promoter reporter activity consistent with impaired WNT/β-catenin signaling. In contrast, WNT/STOP signaling, as determined by c-MYC and cyclin-D1 protein levels was unaffected (Fig. 4e, f and Supplementary Fig. 8). *MYC* and *CCND1* expression were similarly unchanged in *LRP5*-KD cells (Supple-mentary Fig. 7b). Consistent results were obtained in nocodazole-treated cells arrested at G2/M when WNT/STOP signaling is maximal (Supple-mentary Fig. 7c, d). We conclude that LRP5, acting at least partly via WNT/β-catenin signaling, is essential for maintaining the functional properties of APs.

### Mechanisms whereby LRP5 promotes AP fitness

*LRP5*-KD was shown to profoundly suppress growth in mouse mammary epithelial cells due to reduced glucose uptake[40]. WNT-LRP5 signaling was further reported to stimulate osteoblastogenesis by promoting glucose uptake and aerobic glycolysis[41]. Consistent with these reports and the RNA-sequencing data, we observed markedly diminished glucose uptake in *LRP5*-KD APs (Fig. 5a, b). However, supplementing the differentiation media with high-dose pyruvate (5 mM), the three-carbon end-product of glycolysis, failed to rescue adipogenesis (Fig. 5c). We subsequently rescued the expression of *SLC2A3* in abdominal *LRP5*-KD cells, which normalized glucose uptake, but this led only in a small, albeit significant, increase in lipid accumulation during adipogenesis, and failed to reverse the impaired pro-liferation and increased apoptosis of these cells (Fig. 5d–j). Additionally, we assessed the impact of treatment with the GSK3 inhibitor CHIR99021 (0.5 μM). Exposure to CHIR99021 normalized active β-catenin protein

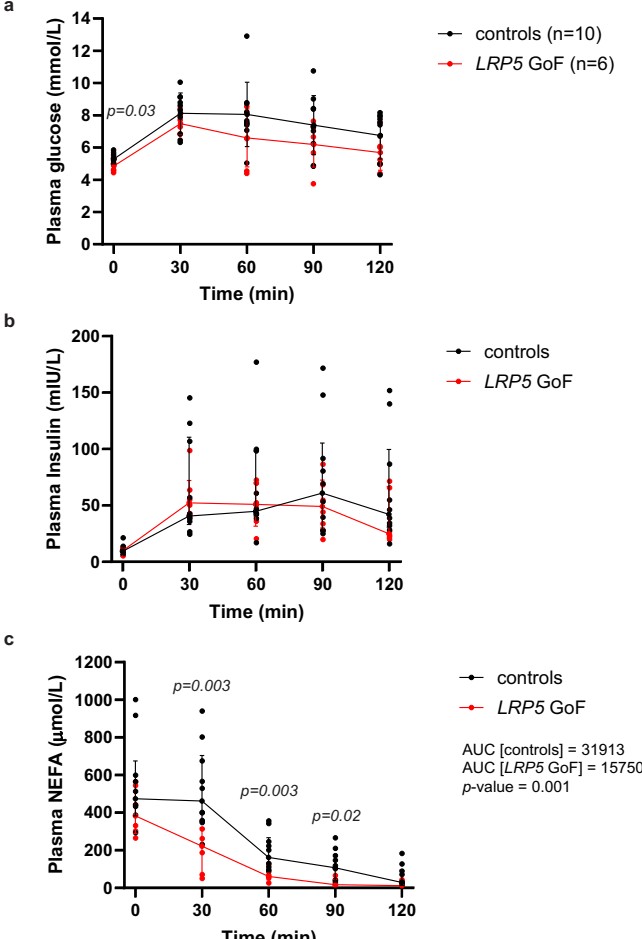

**Fig. 1 | Plasma biochemical profile of subjects with *LRP5* gain-of-function (GoF) mutation and matched controls who underwent oral glucose tolerance tests.** Plasma from blood samples collected at baseline (fasting) and at 30 min intervals (up to 2 h time point) following oral ingestion of glucose was measured for **a** glucose, **b** insulin, and **c** NEFA (median AUC shown). Controls ($n = 10$), *LRP5* GoF ($n = 6$). Graphs are mean ± SD (**a**) and median (IQR) (**b**, **c**). Statistical significance was assessed by two-tailed unpaired Student's *t*-test (**a**) and Mann–Whitney test (**b**, **c**).

**Table 2 | Partial correlations (Spearman's) of measurements of body-fat distribution (DXA), with *LRP5* mRNA levels from abdominal and gluteal adipose tissue fractions from 43 women, adjusted for age, and % total fat mass**

| Traits | SC abdominal APs | | Gluteal APs | | Abdo adipocytes | | Glut adipocytes | |
|---|---|---|---|---|---|---|---|---|
| | $\rho$ | $P$ | $\rho$ | $P$ | $\rho$ | $P$ | $\rho$ | $P$ |
| Android fat mass (g) | −0.143 | 0.4 | −0.271 | 0.09 | 0.134 | 0.4 | 0.049 | 0.8 |
| Gynoid fat mass (g) | 0.390 | 0.01 | 0.285 | 0.08 | 0.108 | 0.5 | 0.196 | 0.2 |
| Leg fat mass (g) | 0.391 | 0.01 | 0.207 | 0.2 | 0.032 | 0.8 | 0.138 | 0.4 |
| Android/gynoid fat ratio | −0.581 | 0.00008 | −0.459 | 0.003 | −0.033 | 0.8 | −0.290 | 0.07 |
| Android/leg fat ratio | −0.511 | 0.001 | −0.386 | 0.01 | −0.032 | 0.8 | −0.209 | 0.2 |

*DXA* dual-energy X-ray absorptiometry, *SC* subcutaneous, *APs* adipose progenitors.

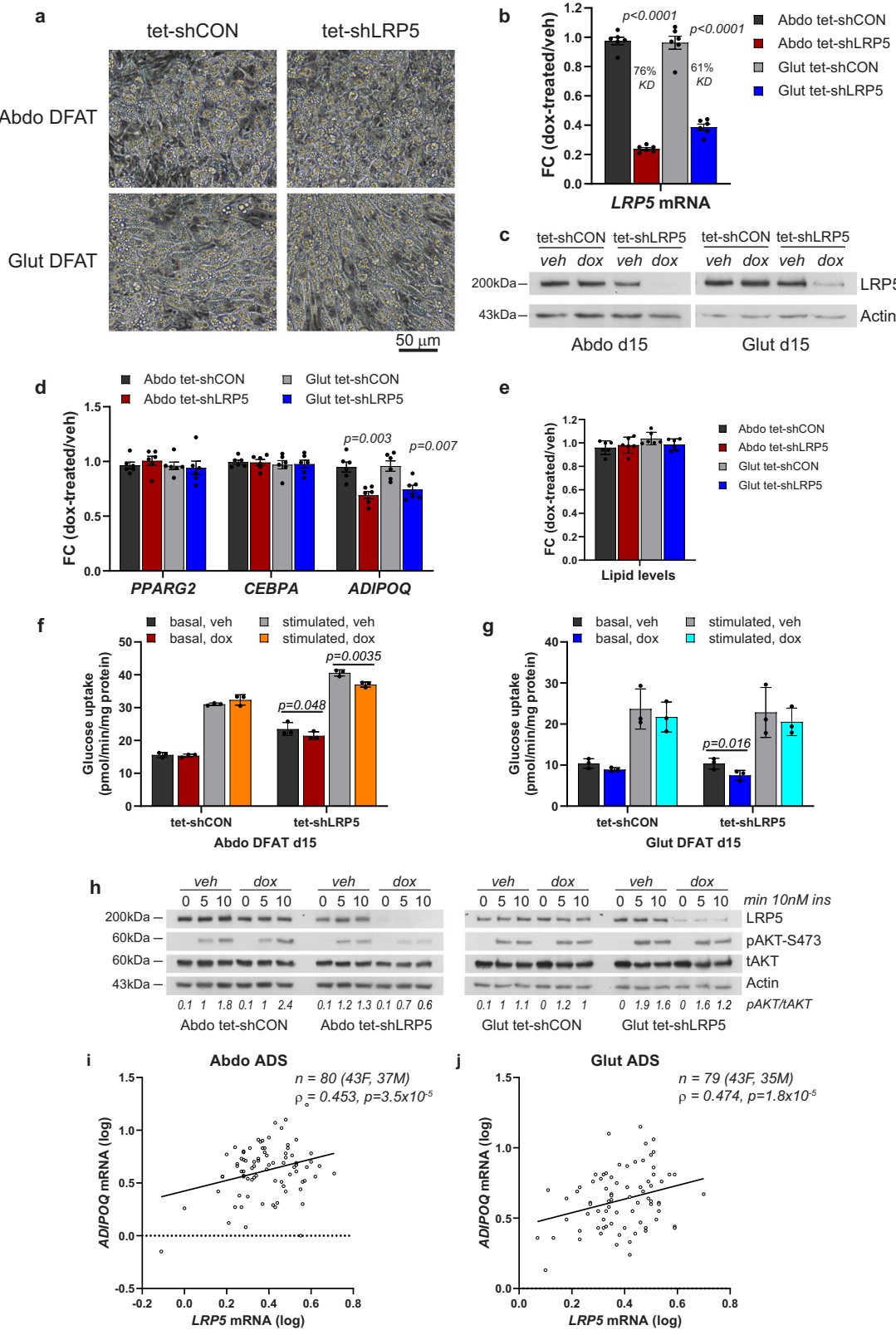

levels in *LRP5*-KD APs but had no effects on c-MYC or cyclin-D1 protein stability (Fig. 5k and Supplementary Figs. 9a–f, 10). At the dose used, CHIR99021 also partially restored proliferation in abdominal *LRP5*-KD APs with a similar trend detected in gluteal cells (Fig. 5l and Supplementary Fig. 9g). Furthermore, CHIR99021 treatment throughout differentiation partially restored lipid accumulation in both abdominal and gluteal APs

(Fig. 5m and Supplementary Fig. 9h). The effects of CHIR99021 on TOP-flash promoter reporter activity and proliferation, but not adipogenesis, where recapitulated by treatment with a WNT surrogate protein (Supplementary Fig. 11) (see Supplementary Methods)[42,43]. In summary, LRP5 promotes AP fitness, at least partly via activation of WNT/β-catenin signaling.

**Fig. 2 | Effects of doxycycline (dox)-induced *LRP5*-knockdown on the expression of adipogenic genes and on basal and stimulated glucose uptake in in-vitro differentiated abdominal and gluteal DFAT cells. a** Light microscopy of abdominal (Abdo) and gluteal (Glut) adipose progenitors (APs) after 12 days of adipogenic differentiation. Scale bar = 50 μm. *LRP5* expression was assessed in DFAT stable cell lines at day 15 of adipogenic differentiation by **b** qRT-PCR (n = 6 experiments; (genotype × dox)$_{Abdo}$ p < 0.0001; (genotype × dox)$_{Glut}$ p = 0.0002) and **c** western blotting, following ~48-h treatment with 0.05 μg/ml doxycycline or vehicle (veh) in hormone-free basal media. **d** qRT-PCR analyses of adipogenic genes *PPARG2*, *CEBPA* and *ADIPOQ* in in-vitro differentiated cells from (**b**) (n = 6; *ADIPOQ*: (genotype × dox)$_{Abdo}$ p = 0.02; (genotype × dox)$_{Glut}$ p = 0.03). **e** Adipogenesis, assessed by AdipoRed staining, was not different between groups (n = 6 replicates). Basal and insulin-stimulated glucose uptake in in-vitro differentiated abdominal (**f**) and gluteal (**g**) DFAT cells following ~48-h treatment with 0.05 μg/ml doxycycline or vehicle in hormone-free basal media (n = 3 independent experiments; (genotype/

insulin × dox)$_{Abdo}$ p = 0.0046; (genotype/insulin × dox)$_{Glut}$ p = 0.22).
**h** Representative western blots showing LRP5 and pAKT-S473 levels in whole cell lysates from day 15 differentiated Abdo and Glut DFAT cells following ~48-h treatment with 0.05 μg/ml doxycycline or vehicle in hormone-free basal media followed by treatment with 10 nM insulin for indicated duration. Correlations between *LRP5* and *ADIPOQ* mRNA levels in isolated mature adipocytes (ADS) from subcutaneous abdominal (**i**) and gluteal (**j**) fat biopsies from 43 females and 37 males. Non-parametric (Spearman's) correlations, adjusted for age, sex and BMI. Statistical significance was assessed by **b**, **d**, **f**, **g** two-way repeated measures ANOVA, and **e** two-way ANOVA, with Sidak's multiple comparisons test comparing doxycycline vs. vehicle-treated groups. qRT-PCR data were normalized to *18S*. Results in (**b**), (**d**), and (**e**) are expressed as fold-change (FC) of dox-treated relative to vehicle-treated samples. **b**, **d**–**g** Histograms are means ± SD. Actin was used as a loading control for western blots.

## Impaired valosin-containing protein function contributes to the phenotype of LRP5 knockdown APs

The top and second most significantly downregulated gene in abdominal and gluteal *LRP5*-KD APs, respectively, was *VCP*, encoding valosin-containing protein (Fig. 3f). VCP protein levels were also lower in *LRP5*-KD APs and *VCP* mRNA abundance in primary abdominal and gluteal APs correlated positively with both *LRP5* gene expression (Fig. 6a–c and Supplementary Fig. 12a) and with lower-body fat distribution (Supplementary Table 9). Furthermore, CHIR99021 treatment failed to prevent the downregulation in *VCP* expression following *LRP5*-KD in APs (Supplementary Fig. 9i). Whilst VCP has multiple functions, one of its main roles is the maintenance of cellular proteostasis by facilitating the degradation of misfolded or damaged proteins through the ubiquitin proteasome system (UPS) and autophagy[44]. We therefore explored if diminished VCP activity might contribute to the impaired fitness of *LRP5*-KD APs. Indeed, induced *VCP*-KD with two independent shRNAs or treatment of abdominal and gluteal DFAT cells with either a competitive (DBeQ) or an allosteric (NMS-873) chemical VCP inhibitor, led to impaired proliferation and adipogenesis independently of WNT/β-catenin signaling. Apoptosis was also increased in *VCP*-KD gluteal APs (Fig. 6d–f and Supplementary Figs. 12b, 13, 14a–c). Next, we explored the effects of VCP depletion on proteostasis. VCP is

important for the maturation of autophagosomes into autolysosomes[45,46]. Consistently, *VCP*-KD for 48- or 96-h led to higher levels of the autophagic substrate LC3-II in gluteal and both abdominal and gluteal APs, respectively, in keeping with defective autophagy (Fig. 6g and Supplementary Figs. 12c, 14d). Additionally, 96-h *VCP*-KD was associated with higher total ubiquitinated protein levels (Fig. 6h and Supplementary Fig. 12d), consistent with impaired proteasomal and/or autophagic protein clearance. Finally, we examined proteostasis in *LRP5*-KD APs. Similar to VCP depletion, 48-h *LRP5*-KD was associated with LC3-II protein accumulation in gluteal cells, whilst *LRP5*-KD for 96-h was associated with higher levels of both LC3-II and total ubiquitinated proteins in both abdominal and gluteal APs (Fig. 6i, j and Supplementary Figs. 12e, f, 14e). In summary, impaired VCP function contributes to the compromised fitness of *LRP5*-KD APs at least partly via defective proteostasis.

## LRP5 and aging-associated fat redistribution

Stem cell exhaustion, disabled autophagy, and loss of proteostasis are hallmarks of aging[47]. Consequently, we investigated if reduced LRP5 function could be linked to aging-associated WAT dysfunction. In fractionated abdominal and gluteal fat biopsies, both AP and adipocyte *LRP5* expression correlated negatively with donor age (Fig. 7a). *LRP5* expression also exhibited the strongest negative correlation with age in WAT among all tissues in GTEx (www.gtexportal.org) (Supplementary Data 2). Furthermore, stable *LRP5*-KD, which was more efficient in gluteal than abdominal APs (72% vs. 26%), was associated with the induction of senescence markers in gluteal APs especially post-induction of differentiation including *MCP-1*, *IL-6*, *CDKN1A*, and *IL1A* (Fig. 7b). Consistently, women carrying GoF *LRP5* variants displayed a lower android-to-leg fat ratio than BMI-matched women who were to 5–20 years younger (Fig. 7c and Supplementary Tables 10, 11). In stark contrast, age- and BMI-matched controls of GoF LRP5 cases exhibited a higher android-to-leg fat ratio than their younger counterparts (Fig. 7c and Supplementary Tables 10, 11). GoF *LRP5* cases were also protected from age-associated bone loss. We conclude that diminished LRP5 activity might contribute to the aging-associated loss of gluteofemoral fat mass consequent to AP functional decline and senescence.

## Discussion

The work described extends our previous study on the role of LRP5 in systemic metabolism and fat distribution[11]. By contrasting the metabolic profiles of GoF and LoF *LRP5* variant carriers, we show that increased LRP5 function is associated with reduced glucose and insulin levels in the fasting state, primarily reflecting hepatic insulin sensitivity. We further show that LRP5 directly promotes adipocyte insulin sensitivity both in vitro and in vivo. In line with these findings, Saarinen et al. reported a high prevalence of type 2 diabetes and impaired glucose tolerance in, mostly heterozygous, carriers of rare LoF *LRP5* mutations[12]. In contrast, another study found no impact of GoF *LRP5* mutations on glucose metabolism[13]. It is worth noting that our GoF *LRP5* cases were more closely and extensively matched with

**Table 3 | Two-sample MR IVW estimates of effects of heel eBMD (g/cm²) on anthropometric and metabolic traits in the European population (sex-combined)**

| Outcome | N SNP | β | SE | p value |
|---|---|---|---|---|
| asatadjbmi3 | 501 | −0.012 | 0.017 | 0.49 |
| gfatadjbmi3 | 501 | −0.102 | 0.020 | $6.41 \times 10^{-7}$ |
| vatadjbmi3 | 501 | 0.009 | 0.020 | 0.65 |
| BMI | 498 | 0.015 | 0.009 | 0.077 |
| WHRadjBMI | 498 | 0.007 | 0.011 | 0.54 |
| Waist circumference (adjBMI) | 368 | −0.019 | 0.015 | 0.20 |
| Hip circumference (adjBMI) | 368 | −0.048 | 0.018 | 0.0084 |
| Fasting glucose (adjBMI) | 497 | 0.004 | 0.005 | 0.45 |
| Fasting insulin (adjBMI) | 499 | 0.014 | 0.006 | 0.011 |
| logTAG_noUKB | 500 | 0.012 | 0.009 | 0.15 |
| HDL_noUKB | 499 | −0.019 | 0.010 | 0.050 |
| LDL_noUKB | 499 | −0.012 | 0.013 | 0.36 |

*asatadjbmi3* BMI and height-adjusted abdominal subcutaneous adipose tissue, *BMI* body mass index, *gfatadjbmi3* BMI and height-adjusted gluteofemoral adipose tissue, *eBMD* estimated bone mineral density, *HDL* high-density lipoprotein, *LDL* low-density lipoprotein, *TAG* triglycerides, *vatadjbmi3* BMI and height-adjusted visceral adipose tissue, *WHRadjBMI* BMI-adjusted waist-to-hip ratio.

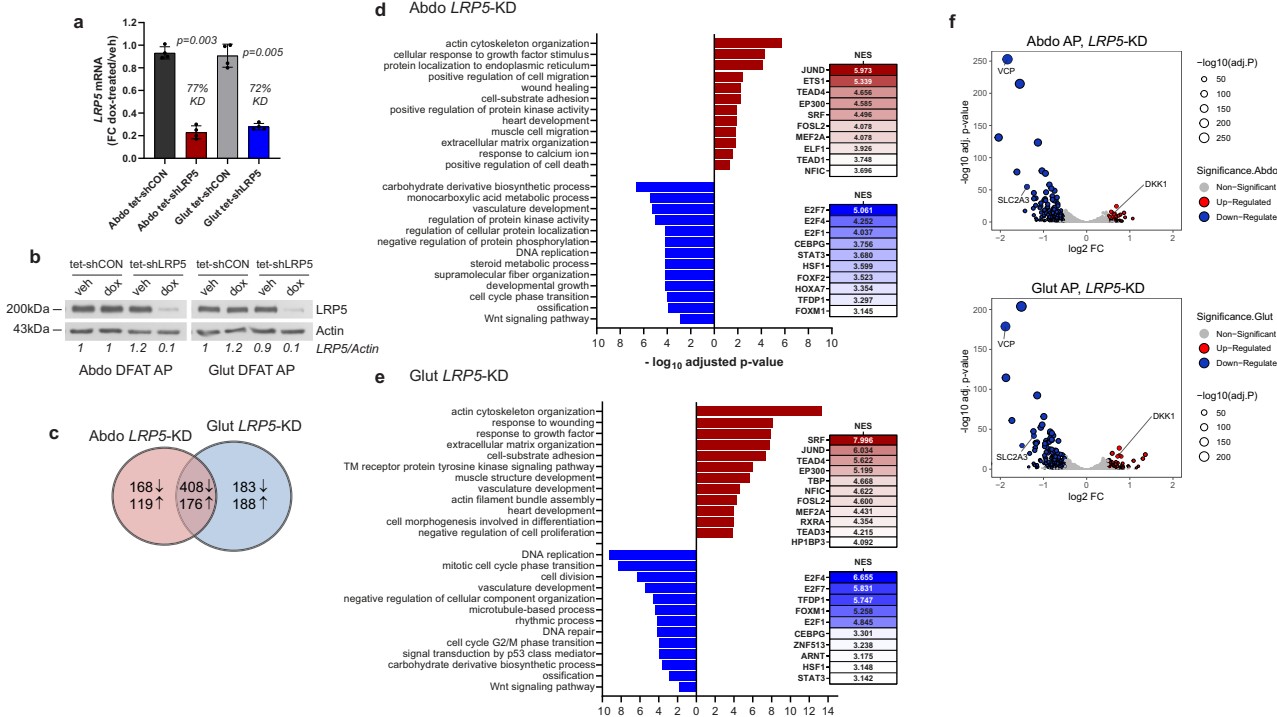

**Fig. 3 | Global transcriptional profiling reveals that LRP5 regulates multiple aspects of adipose progenitor (AP) biology. a, b** Abdominal (Abdo) and gluteal (Glut) DFAT APs, stably transduced with the tetracycline-inducible control (tet-shCON) or shLRP5 (tet-shLRP5) vector were cultured in the presence of vehicle (veh) or doxycycline (dox) (final concentration of 0.1 μg/ml) for 48 h to induce shRNA expression. *LRP5*-knockdown (KD) was confirmed by **a** qRT-PCR (*n* = 4 experiments) (genotype × dox)$_{Abdo}$ *p* = 0.006; (genotype × dox)$_{Glut}$ *p* = 0.01, and **b** western blot. qRT-PCR data were normalized to *18S* and expressed as fold-change (FC) gene expression of dox-treated samples relative to vehicle-treated samples. Histograms are means ± SD. Statistical significance was assessed by two-way

repeated measures ANOVA, with Sidak's multiple comparisons test comparing doxycycline vs. vehicle-treated groups. **c** Venn diagram showing the number of significantly (padj < 0.05) up- and downregulated genes with doxycycline-induced *LRP5*-KD. Pathway enrichment analyses of genes upregulated (red) and downregulated (blue) with doxycycline-induced *LRP5*-KD in **d** abdominal and **e** gluteal APs. Results of transcription factor binding-site motif analysis of differentially expressed genes, with normalized enrichment scores (NES), are shown to the right. **f** Volcano plots of genes with padj < 0.05 and log$_2$ fold-change (log$_2$ FC) > 0.5 in red (upregulated) or blue (downregulated). Three genes investigated in this study are labeled. Actin was used as a loading control for western blots.

controls. Additionally, our conclusions are strengthened by the contrasting metabolic phenotypes of GoF and LoF *LRP5* variant carriers.

We previously showed that GoF *LRP5* mutations were associated with lower android-to-leg fat ratio[11]. Here, we confirm and expand this finding by showing that the favorable fat distribution in LRP5 GoF variant carriers is primarily driven by higher leg fat mass. Our earlier work also identified a common, intronic *LRP5* SNV (rs599083) linked to low spinal BMD and a nominally increased android-to-leg fat ratio[11]. However, the causal gene(s) and cell type(s) responsible for the associations with the bone and potentially the adipose phenotypes at this signal remain unknown. Therefore, we revisited this finding by examining the fat distribution of carriers of LRP5$_{V667M}$, a fine mapped, missense *LRP5* variant, predicted to be moderately pathogenic[48], which was associated with reduced heel eBMD in GWAS[24]. Similar to humans, homozygous LRP5$_{V667M}$ mutant mice displayed reduced BMD[49] and LRP5$_{V667M}$ was further shown to be associated with impaired WNT reporter activation[50]. Contrasting the phenotype of GoF *LRP5* cases, LRP5$_{V667M}$ variant carriers exhibited reduced leg fat mass.

Animal studies have revealed that LRP5 can influence adiposity and systemic metabolism in a cell non-autonomous manner. Mice lacking *Lrp5* in mature osteoblasts and osteocytes displayed decreased postnatal bone mass alongside higher fat mass, elevated plasma triglycerides and NEFA on a chow diet, and glucose intolerance and insulin resistance following a HFD[51,52]. Conversely, animals expressing a HBM *Lrp5* mutant allele in osteoblasts and osteocytes displayed the opposite phenotype[51]. However, MR studies conducted here revealed a negative impact of higher heel eBMD on lower-body fat mass and potentially insulin sensitivity. These findings indicate that the effects of the functional LRP5 variants on these traits in

humans are probably dissociated from their actions in bone. Consistently, in vitro assays demonstrated that LRP5 could cell autonomously regulate adipocyte glucose uptake, insulin signaling, and *ADIPOQ* mRNA abundance. In this context, hypoadiponectinaemia is a marker of post-receptor adipocyte insulin resistance[53]. The mechanism(s) whereby LRP5 promotes insulin action in human adipocytes remains unclear. Earlier work showed that the insulin receptor and LRP5 interact in both an insulin and WNT inducible manner in 3T3-L1 preadipocytes[8]. In contrast, another study demonstrated that *Lrp5*-deficient osteoblasts were insulin resistant due to intracellular accumulation of diacylglycerol species[52]. We speculate (see below), that hyperplastic subcutaneous WAT expansion also contributes to the enhanced adipocyte insulin sensitivity associated with GoF *LRP5* mutations. Finally, the lack of a reduction in insulin-stimulated glucose uptake in LRP5-depleted gluteal adipocytes indicates that either LRP5 does not modulate insulin signaling in gluteal adipocytes or, alternatively, that abdominal and gluteal adipocytes have differential insulin sensitivity, possibly with the dose of insulin used (10 nmol/L) obscuring the effect of LRP5 depletion in gluteal cells.

Based on data showing differential LRP5 gene and protein expression specifically in the AP fraction of abdominal vs. gluteal WAT (abdominal > gluteal), and that stable and equivalent *LRP5*-KD in APs led to more pronounced defects in proliferation and adipogenesis in gluteal cells, we previously suggested that APs mediate the favorable effects of LRP5 on fat distribution[11]. In line with this hypothesis, we now demonstrate that AP *LRP5* expression correlates selectively and positively with lower-body fat mass. Analysis of RNA-sequencing data from *LRP5*-KD DFAT cells also provided evidence that LRP5 stimulates both osteoblastogenesis and

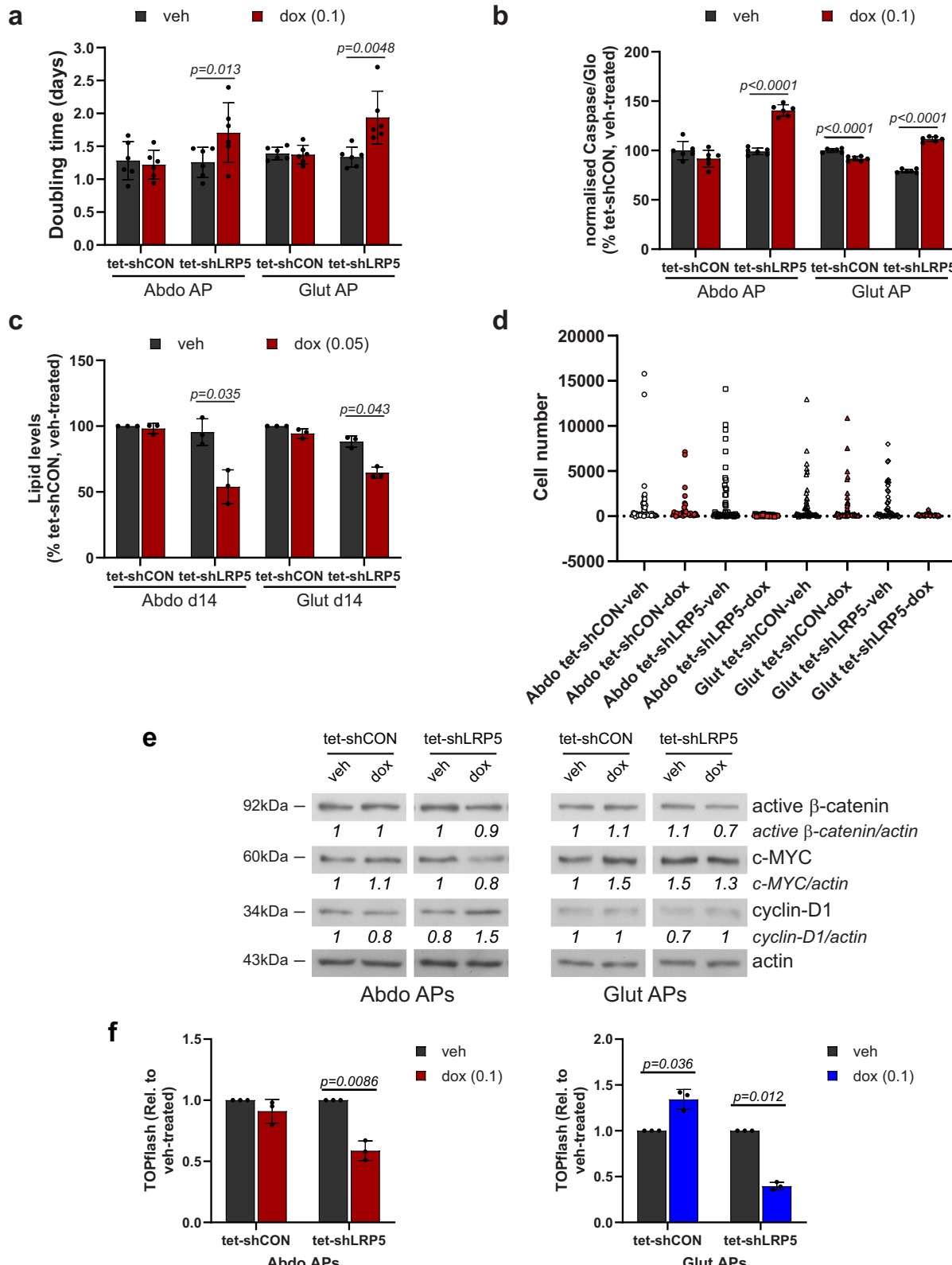

**Fig. 4 | Effects of doxycycline-induced *LRP5*-knockdown (KD) on abdominal (Abdo) and gluteal (Glut) adipose progenitor (AP) biology.** Effects of doxycycline (dox)-induced *LRP5*-KD on: **a** doubling time ($n = 6$ independent experiments; (genotype × dox)$_{Abdo}$ $p = 0.02$; (genotype × dox)$_{Glut}$ $p = 0.009$); **b** apoptosis ($n = 6$, representative of two independent experiments. Results were normalized to cell number. (genotype × dox)$_{Abdo}$ $p < 0.0001$; (genotype × dox)$_{Glut}$ $p < 0.0001$); **c** adipogenesis ($n = 3$ independent experiments, (genotype × dox)$_{Abdo}$ $p = 0.04$; (genotype × dox)$_{Glut}$ $p = 0.07$); and **d** clonogenic potential ($n = 48$ clones/group, representative of two independent

experiments). **e** Western blots of active β-catenin, c-MYC and cyclin-D1 in vehicle (veh) and doxycycline-treated APs. **f** Normalized TOPflash activity in tet-shCON/7TFC and tet-shLRP5/7TFC cells treated with vehicle or 0.1 µg/ml doxycycline for 96 h ($n = 4$ independent experiments, (genotype × dox)$_{Abdo}$ $p = 0.01$; (genotype × dox)$_{Glut}$ $p = 0.005$). **a–c, f** Histograms are means ± SD. Statistical significance was assessed by **a, c, f** two-way repeated measures ANOVA, with Sidak's multiple comparisons test, and **b** two-way ANOVA with Sidak's multiple comparisons test. Actin was used as a loading control for western blots.

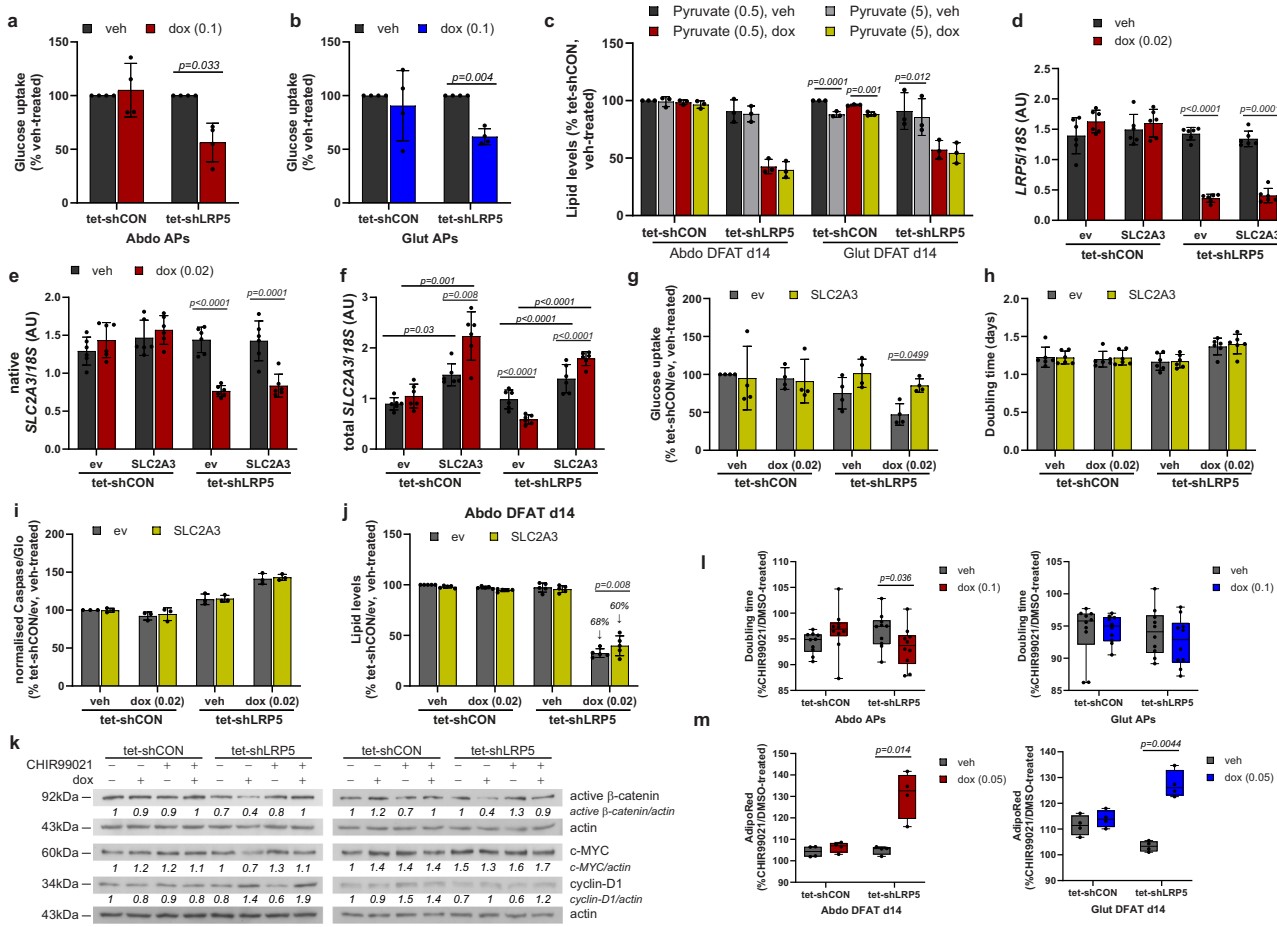

**Fig. 5 | Restoration of *SLC2A3*-expression and treatment with a GSK3 inhibitor, CHIR99021, are not able to prevent defects in DFAT cells due to *LRP5*-knockdown (KD). a**, **b** Abdominal (Abdo) and gluteal (Glut) DFAT adipose progenitors (APs) with doxycycline (dox)-induced *LRP5*-KD have reduced basal glucose uptake (*n* = 4 independent experiments; (genotype × dox)Abdo *p* = 0.03; (genotype × dox)Glut *p* = 0.17). **c** Sodium pyruvate (5 mM) supplementation is not able to rescue adipogenesis in *LRP5*-KD cells (*n* = 3 independent experiments; (genotype/dox × pyruvate)Abdo *p* = 0.49; (genotype/dox × pyruvate)Glut *p* = 0.005). **d**–**j** *SLC2A3* (GLUT3)-overexpression in abdominal *LRP5*-KD APs increases basal glucose uptake and partially rescues adipogenesis. Dox-induced overexpression of *SLC2A3* in control and *LRP5*-KD abdominal DFAT cell lines. mRNA levels of **d** *LRP5*, **e** native *SLC2A3*, and **f** total *SLC2A3* [(SLC2A3 × dox)Abdo tet-shCON *p* = 0.02, (SLC2A3 × dox)Abdo tet-shLRP5 *p* < 0.0001] were assessed by Taqman PCR (*n* = 6 experiments). qPCR data were normalized to *18S*. Effects of *SLC2A3*-overexpression in abdominal DFAT APs on: **g** basal glucose uptake (*n* = 4 independent experiments, (genotype/dox × SLC2A3)Abdo *p* = 0.084), **h** doubling time (*n* = 6 independent experiments, (genotype/dox × SLC2A3)Abdo *p* = 0.28), **i** apoptosis (*n* = 3 independent experiments; (genotype/dox × SLC2A3)Abdo *p* = 0.95), and **j** adipogenesis

(*n* = 5 independent experiments; (genotype/dox × SLC2A3)Abdo *p* = 0.008). Impaired adipogenesis due to *LRP5*-KD was less severe in cells overexpressing SLC2A3 vs. empty vector (60% vs. 68% reduction, relative to vehicle-treated tet-shCON cells). **k**–**m** Treatment with a GSK3 inhibitor, CHIR99021, partially rescues inhibition in proliferation and adipogenesis due to *LRP5*-KD in Abdo and Glut DFAT APs. **k** Representative western blots showing the effects of 0.5 μM CHIR99021 (CHIR) treatment on active β-catenin, c-MYC and cyclin-D1. Effects of CHIR99021 treatment on: **l** doubling time (*n* = 10 independent experiments; (genotype × dox)Abdo *p* = 0.006; (genotype × dox)Glut *p* = 0.34), and **m** adipogenesis (*n* = 4 independent experiments; (genotype × dox)Abdo *p* = 0.02; (genotype × dox)Glut *p* = 0.008). Graphs are shown as a fold-change (fc) of CHIR99021-treated vs. DMSO-treated cells. In (**m**), cells were treated with CHIR99021 (or vehicle) throughout differentiation. See also Supplementary Fig. 9. Statistical tests: **a**–**c**, **g**–**j**, **l**, **m** two-way repeated measures ANOVA, with Sidak's multiple comparisons test; **d**–**f** two-way repeated measures ANOVA with Tukey's multiple comparisons test. Histograms are means ± SD. Box and whisker plot: Whiskers are maximum and minimum values, and box represents median and interquartile range. Actin was used as a loading control for western blots.

adipogenesis in APs, further arguing that the bone and adipose phenotypes associated with LRP5 variants are primarily driven by LRP5 actions on mesenchymal stem cell biology. The transcriptomic analyses additionally supported a role for LRP5 in stimulating AP proliferation, as well as processes which are essential for cellular energy production and biomass accumulation in proliferating cells, namely glucose uptake, glycolysis, and de novo lipogenesis. It is also noteworthy that the promoters of DEGs in *LRP5*-KD APs were enriched for motifs of several TFs previously shown to be induced during the mitotic clonal expansion of preadipocyte cell lines and to modulate adipogenesis, including members of the AP1 (FOS, JUN)[54], ETS[55] and E2F[56] families. This finding along with the temporal effects of *LRP5*-KD on adipogenesis reported here suggest that LRP5 controls in vivo

adipogenesis, at least partly by promoting AP self-renewal, which precedes differentiation of one of the two resulting daughter cells[57]. Consistently, it was recently reported that transient WNT signaling activation upon in vitro adipogenic induction is important in maintaining a pool of multipotent progenitors[58,59].

Functional studies supported and extended the results of the transcriptomic analyses, by demonstrating that LRP5 depletion might limit subcutaneous WAT expansion by compromising the fitness of APs; namely their renewal, proliferation, survival, and adipogenic potential. In these experiments, *LRP5*-KD did not result in depot-dependent effects on AP function, consistent with the near total *LRP5*-KD achieved in abdominal and gluteal DFAT cells using an inducible shRNA system here, as opposed

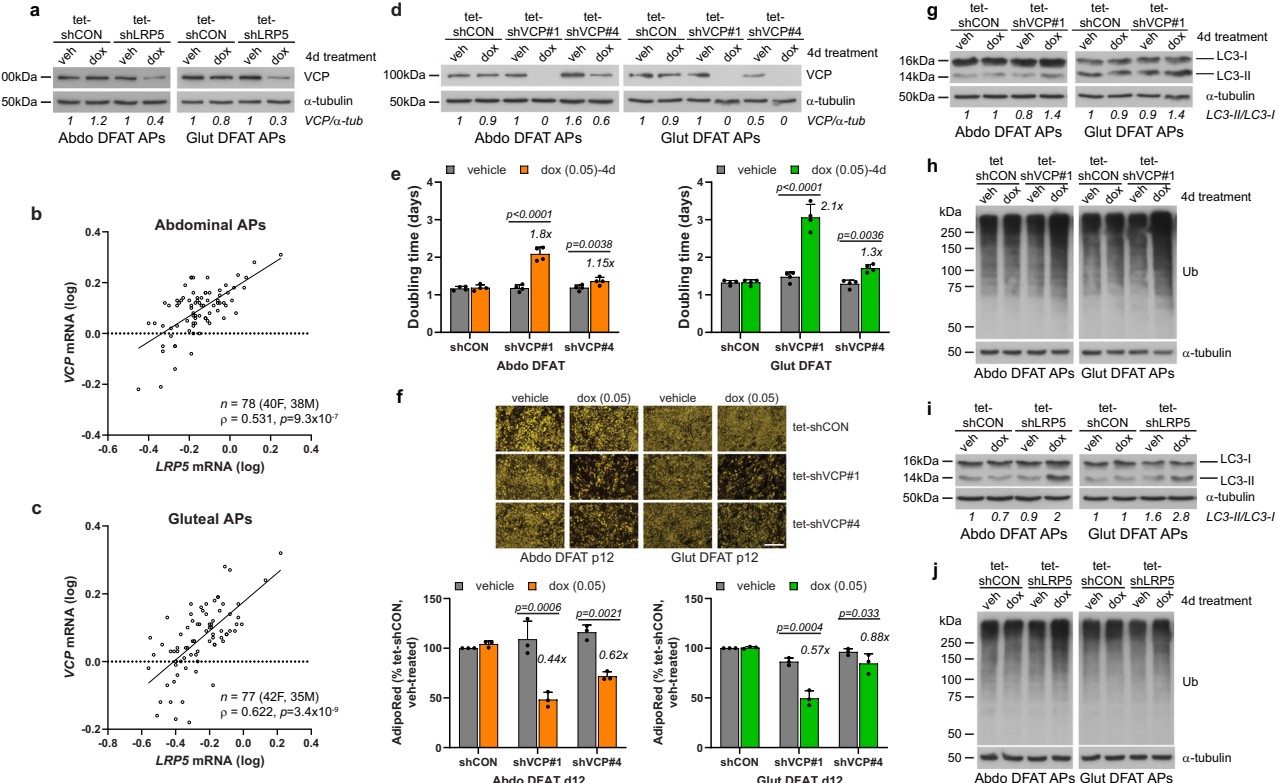

**Fig. 6 | *LRP5*-knockdown (KD), VCP protein expression, and proteostasis.**
**a** Western blot of VCP in tet-shCON and tet-shLRP5 abdominal (Abdo) and gluteal (Glut) DFAT adipose progenitors (APs) following 4 days treatment with vehicle (veh) or 0.1 μg/ml doxycycline (dox). Correlations between *LRP5* and *VCP* mRNA expression levels in human cultured primary APs from subcutaneous abdominal (**b**) and gluteal (**c**) fat biopsies from 40 to 42 females and 35 to 38 males. Non-parametric (Spearman's) correlations, adjusted for age, sex and BMI. qRT-PCR results are normalized to *18S*. **d–h** Effects of doxycycline-induced *VCP*-KD on DFAT AP biology. **d** Western blots showing *VCP*-KD by two independent shRNAs (shVCP#1 and shVCP#4) following treatment with 0.05 μg/ml doxycycline for 4 days. Effects of doxycycline-induced *LRP5*-KD on: **e** doubling time ($n = 4$ independent experiments;

(genotype × dox)$_{Abdo}$ $p < 0.0001$; (genotype × dox)$_{Glut}$ $p < 0.0001$; **f** adipogenesis (vehicle or dox-treatment throughout adipogenesis, $n = 3$ independent experiments; (genotype × dox)$_{Abdo}$ $p = 0.0013$; (genotype × dox)$_{Glut}$ $p = 0.0013$). Scale bar = 200 μm; **g** autophagy (indicated by autophagic marker LC3-II); and **h** ubiquitinated protein (Ub) levels. Cells were treated for 4 days with 0.05 μg/ml doxycycline or vehicle prior to assay in (**d**, **e**, **g** and **h**), and throughout adipogenesis in (**f**). Effects of 4-day doxycycline-induced *LRP5*-KD on autophagy (indicated by autophagic marker LC3-II) (**i**), and ubiquitinated protein (Ub) levels (**j**), in DFAT APs. Statistical analyses: **e**, **f** two-way repeated measures ANOVA with Sidak's multiple comparisons test. Histograms are means + SD. α-tubulin was used as a loading control for western blots.

to a stable system in our previous study[11]. Kato et al. similarly showed that the low bone mass of global *Lrp5*-deficient mice was secondary to osteoblast defects[60]. Furthermore, *Lrp5* deficiency led to impaired mammary stem cell maintenance[61]. Collectively these studies and our own data indicate that LRP5 might promote progenitor cell fitness in diverse tissues.

Mechanistically, *LRP5*-KD in APs was associated with impaired WNT/β-catenin but not WNT/STOP pathway activity arguing that the former is the dominant pathway in APs. Accordingly, WNT/β-catenin signaling activation with low-dose CHIR99021 partially rescued the impaired proliferation and, unexpectedly, the defective lipid accumulation during adipogenesis in *LRP5*-KD cells. In this regard, WNT/β-catenin signaling has been shown to potently inhibit adipogenesis[62]. Based on our current, as well as previous findings[11,17], we speculate that this pathway has dose-dependent effects on adipogenesis, with mild activation insufficient to block and possibly stimulating early differentiation, in addition to promoting lipid accumulation via de novo lipogenesis during the latter stages of adipogenesis[63,64]. We further hypothesize that the differences in rescue efficacy between abdominal and gluteal *LRP5*-KD APs are due to differential sensitivity to CHIR99021. We observed that a WNT surrogate partially rescued proliferation in LRP5-depleted APs but, in contrast to CHIR99021, further impaired differentiation. These discrepancies could be due to functional redundancy between LRP5 and LRP6[65], over-stimulation of WNT/β-catenin signaling with the WNT surrogate at the doses used, or alternatively, the independence of the effects of CHIR99021 on adipogenesis

from WNT/β-catenin signaling. Surprisingly, supplementation with pyruvate or rescue of glucose uptake failed to improve AP fitness in *LRP5*-KD cells. These data argue that in high glucose culture conditions (17.5 mM), glucose uptake and glycolysis might not be limiting for growth and differentiation of *LRP5*-KD cells. Alternatively, impaired glucose uptake/metabolism might be a consequence rather than a driver of this phenotype in DFAT cells.

Our study also shows that impaired VCP activity contributes to the compromised function of *LRP5*-KD cells. Specifically, *LRP5* and *VCP* expression were strongly and positively correlated in primary APs and *LRP5*-KD in DFAT APs was associated with a striking reduction in VCP gene and protein levels. Furthermore, both genetic and chemical inhibition of VCP function replicated the phenotype of *LRP5*-KD cells. Finally, *LRP5*-KD APs displayed defective proteostasis, consistent with the key role of VCP in the UPS and autophagy. The ability to maintain a functional proteome by clearing, damaged or misfolded proteins is critical for maintaining cell fitness and survival. Accordingly, mutations in VCP cause multisystem proteinopathies, characterized by degeneration of bone and muscle, i.e., mesenchymal tissues, as well as the brain[44]. We speculate that these patients also have WAT dysfunction. Our results also suggest that altered VCP function and proteostasis contribute to the adipose and metabolic phenotypes associated with functional LRP5 variants. VCP was also shown to be involved in other key cellular processes including cell cycle progression, genomic stability and endosomal sorting[44], which could also contribute to

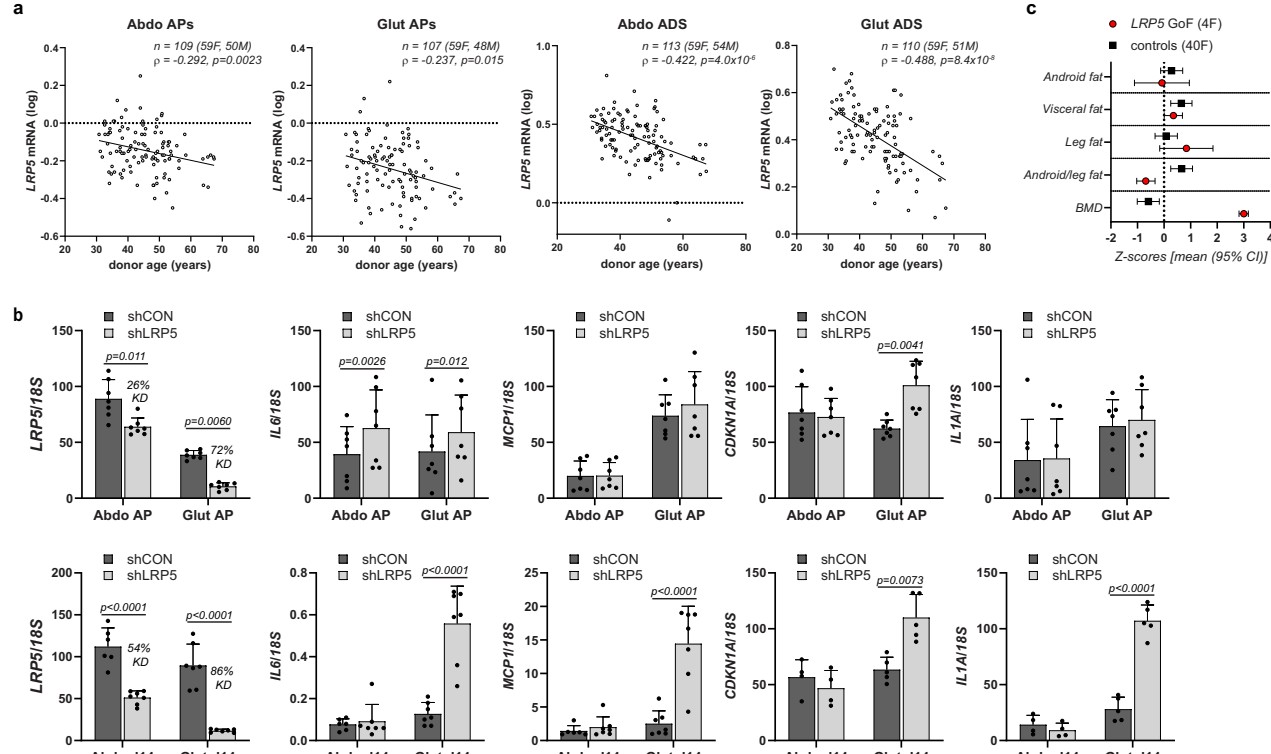

**Fig. 7 | LRP5 and aging. a** *LRP5* mRNA expression in human cultured primary adipose progenitors (APs) and isolated adipocytes is reduced with donor age. Correlations between *LRP5* mRNA levels in abdominal (Abdo) and gluteal (Glut) APs and adipocytes (ADS), and donor age. Non-parametric (Spearman's) correlations, adjusted for sex and BMI. **b** Expression of senescence markers is increased with constitutive *LRP5*-knockdown (KD) in undifferentiated and in vitro differentiated immortalized APs. shCON = control cells, shLRP5 = *LRP5*-KD cells. (experimental replicates: AP cells, *n* = 7; d14 cells: for *LRP5*, *MCP1* and *IL6* qRT-PCR, *n* = 7 [*n* = 6 for Abdo shCON], for *CDKN1A* and *IL1A* qRT-PCR: *n* = 4 for Abdo d14 and *n* = 5

for Glut d14). Statistics: two-way repeated measures ANOVA (for APs) and mixed-effects analysis (d14 cells), with Sidak's multiple comparisons test. qRT-PCR data were normalized to *18S*. Histograms are means + SD. **c** Female *LRP5* gain-of-function (GoF) mutation carriers are protected against age-related lower-body fat loss. Comparison of body composition (DXA) of women with *LRP5* GoF (A242T, N198S) variants and matched controls within a cluster of sex- and BMI-matched individuals 5–20 years younger (see also Supplementary Tables 10 and 11). BMD bone mineral density.

the compromised functionality of *LRP5*-KD cells. How LRP5 regulates VCP expression remains another important question.

During aging, there is a progressive loss of peripheral WAT in the legs and arms coupled with the accumulation of visceral fat[66]. Consistently, the proliferation and differentiation capacities of APs decline during aging in both humans[66,67] and mice[68]. It was further shown that subcutaneous AP numbers dramatically decline with age in old mice[68]. Our data demonstrate that reduced *LRP5* expression in APs might play a key role in the WAT redistribution and dysfunction with advancing age by leading to AP dysfunction and senescence. In line with these findings, female carriers of GoF *LRP5* mutations were protected from the aging-associated loss of lower-body fat. Furthermore, because LRP5 expression is lower in gluteal APs (Fig. 7b and[11]), it is likely that the gluteofemoral depot is more vulnerable to perturbations of LRP5 function through missense variants or gene expression changes. Consistently, stable *LRP5*-KD in APs was associated with increased expression of senescence-associated markers selectively in gluteal APs.

In summary, we demonstrate that LRP5 promotes systemic and adipocyte insulin sensitivity. Additionally, LRP5 plays a critical role in promoting a lower-body fat distribution by maintaining the functional characteristics of APs, at least partly through WNT/β-catenin signaling activation and independently of this pathway by preserving proteostasis via VCP activity. *LRP5* expression in APs and WAT declines with age and accordingly, *LRP5* GoF mutation carriers were protected from the lower-body fat loss associated with aging, which we propose is the osteoporosis equivalent in WAT. Pharmacologic activation of LRP5 in WAT offers a

promising approach to ameliorate the fat redistribution, metabolic complications and bone loss associated with aging.

## Data availability

Source data for Figs. 1–7 and gene counts for RNA sequencing data are available in Supplementary Data 3. Other data and resource generated during the current study are available from the corresponding author upon reasonable request. All MR analyses were conducted using publicly available data with links to GWAS sources available in Supplementary Table 2.

## Code availability

All analyses were completed with existing software packages.

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

## Acknowledgements

We are grateful to the OBB and HBM volunteers, and the Clinical Research Unit nurses for their help in recruiting volunteers and with sample collection, to Dr. Toryn Poolman and Prof. Johnathan Labbadia for their advice regarding the R package and the proteostasis experiments, respectively, to Dr Enrique Toledo Maldonado for his help with scRNA-seq studies, and to Prof. Clive Osmond for advice on statistics. This work was funded by a British Heart Foundation Clinical Research Fellowship to C.C. (FS/16/45/32359). The OBB and Oxford BioResource are funded by the NIHR Oxford BRC. F.K. was funded by a BHF program grant (RG/17/1/32663). We would also like to acknowledge funding support from the NIHR Oxford BRC, Diabetes & Metabolism Theme (IS-BRC-1215-20008), and the European Foundation for the Study of Diabetes. C.L.G. received funding for the HBM study from The Wellcome Trust (080280/Z/06/Z) and Versus Arthritis (20000). The views expressed are those of the authors. C.C. is the guarantor of this work, as such, has full access to all the study data, and takes responsibility for the integrity of the data and the accuracy of the data analysis.

## Author contributions

Conceptualization, C.C.; methodology, N.Y.L., C.C. and E.R.; investigation, N.Y.L., C.C., E.R., A.D.vD. and M.V.; formal analysis, N.Y.L., S.K.V., D.B.R., E.R., A.D.vD., D.P., A.W-A., M.J.N. and R.N.; visualization, N.Y.L., E.R., A.D.vD. and D.P.; writing—original draft, N.Y.L. and C.C.; writing—review and editing, N.Y.L., S.K.V., D.B.R., E.R., A.D.vD, M.V., D.P., A.W-A., M.J.N., R.N., D.W.R., J.H.T., C.L.G., F.K. and C.C.; funding acquisition, C.C. Resources, C.C., F.K., C.L.G. and J.H.T.; supervision, C.C.

## Competing interests
