## [Peer review file · Communications Medicine]

LRP5 promotes adipose progenitor cell fitness and adipocyte insulin sensitivity.

Corresponding Author: Dr Constantinos Christodoulides

Version 0:

Reviewer comments:

Reviewer #1

(Remarks to the Author)

In the manuscript "LRP5 promotes adipose progenitor cell fitness and adipocyte insulin sensitivity", Loh et al. investigate the role of LRP5 in systemic metabolism, as well as the cellular and molecular mechanisms that underlie the impact of LRP5 on fat distribution. For this purpose, the authors studied gain-of-function and loss-of-function LRP5 variant carriers and performed functional assays in human adipocytes and precursors. The authors provide evidence that LRP5 boosts insulin sensitivity both systemically and within adipocytes. LRP5 plays a vital role in promoting a specific pattern of fat distribution favoring the lower body. It achieves this by preserving the functional traits of adipocytes, partly through activating WNT/ β -catenin signaling and also by maintaining proteostasis via VCP activity, independent of the WNT/ β -catenin pathway. Interestingly, individuals with LRP5 gain-of-function (GoF) mutations were shielded from the typical age-related loss of lower-body fat. The manuscript is well written, and easy to read. The scientific background and rationale for the investigation is clearly addressed. The work is rigorously performed, and interpretation of data is appropriate. This group is a world-leader in this area.

Minor

- 1) Can the authors speculate how the depletion of LRP5 causes a decrease in basal glucose uptake in gluteal adipocytes and a decrease in stimulated glucose uptake in abdominal adipocytes?
- 2) As a control, it might have been interesting to use purified recombinant Wnts as additional treatments to Chir99021 to evaluate [presumably lack of] functionality of the cells.
- 3) For future studies, it would be interesting to isolate cells from adipose tissue of patients and explore directly the functional effects of LRP5 variants.

Reviewer #2

(Remarks to the Author)

Comments to the manuscript COMMSMED-24-0196-T:

The manuscript entitled "LRP5 promotes adipose progenitor cell fitness and adipocyte insulin sensitivity" by Loh NY et al., by studying LRP5 gain- and loss-of-function adipose and metabolic phenotypes, shows that LRP5 promotes lower-body fat distribution and enhances systemic and adipocytes insulin sensitivity. The authors also demonstrate via Mendelian Randomisation analysis and functional studies in LRP5 KD adipocytes, that LRP5's effects on these traits are mostly cell autonomous and independent of its impact on bones. LRP5 promotes lower-body fat distribution by preserving the functional characteristics of adipose progenitors, partly through WNT/ β -catenin pathway activation, and independently by preserving valosin-containing protein expression and proteostasis. Moreover, both adipocyte and AP LRP5 expression decline with age. Accordingly, gain-of-function LRP5 variant carriers are protected against the age-associated loss of lower-body fat.

Major comments

p.8 line 227. The authors mention "Consistent with these findings, subcutaneous WAT LRP expression...". The subcutaneous adipose tissue here is the gluteal one? The authors should indicate that in the text. Same for line 234, p.9, the subcutaneous adipose tissue from where is from? lower body? Gluteal?

Fig.2F. The error bars in this panel are SD or SEM?

Fig.2H. The authors should provide quantification of the WB.

Fig.3B. The authors should provide quantification of the WB.

Fig.4E. The authors should provide quantification of the WB.

Fig.5K. The authors should provide quantification of the WB.

Fig.6A, D, G and I. The authors should provide quantification of the WBs.

Reviewer #3

(Remarks to the Author)

The paper nicely combines experimental evidence with evidence from human genetics utilising Mendelian randomisation.

Focus on the latter:

1. The authors describe selecting instruments using a p-value of 5×10^{-8} as well as an F-statistic of 24. This is incorrect, the F-statistic and p-value are proportional to each other and the mentioned threshold are inconsistent. The aforementioned p-value is equal to an F-statistic of about 29, and the mentioned F-statistic of 24 is equal to a p-value of about 1×10^{-6} . Please fix this and simply mention either the F-statistic or p-value, but not both.
2. It is not entirely clear but presumably the authors attempted to conduct a cis MR study selecting variants from within and around LRP5? Please clarify and consider referring to the design as a cis-MR study:
<https://www.nature.com/articles/s41467-020-16969-0> .
3. Assuming this was a cis-MR why did the authors use a 10Mpb flanking region? Presumably such a large region would include a considerable number of other genes (e.g., about 10 or so) which may give rise to the associations described. Hence the current results may not represent the effects of LRP5. Please either explain why such a large flanking region is used, or consider using a smaller flank (e.g., 100 kbp +/- LRP5).
4. Which minor allele frequency was used?
5. Which data were used for the LD calculations and do the authors consider these data sufficiently large to calculate LD for the minor allele frequency chosen (whichever that might be).
6. "An IVW result with $p < 0.05$, coupled with directionally consistent associations in both sensitivity analyses was considered evidence for causal association." This is a rather liberal selection strategy, would be good to also identify analysis significant in all of the MR analysis, as well as 2 out of 3 (assuming the total number of MR estimator used was 3)?
7. Please add units of the exposure when discussing the MR results, as well as in the tables and other display items.
8. Please use scientific notation instead of e-notation throughout.
9. The authors describe using a two-sample MR design. However going a quick look at the manuscript suggest some of them have included the UKB such a Pulit or Chen (which is the same as used for the exposures). Please consider adding a table with the GWAS studies used and whether they included the UKB or not. If they included the UKB please try to identify the sample overlap.

As a complete aside "Statistical analyses were performed using R, Stata, SPSS and/or GraphPad" using 4 statistic software packages is a bit much – 2 should have been sufficient.

Version 1:

Reviewer comments:

Reviewer #1

(Remarks to the Author)

Thank you - this is a terrific paper.

Reviewer #2

(Remarks to the Author)

We are satisfied with how the authors addressed our comments. We consider now the manuscript suitable for publication in Communications Medecine.

Reviewer #3

(Remarks to the Author)

Most of my questions have been addressed, although in future it would be beneficial to indicate in the cover letter which changes were made in to manuscript. The following comments remain:

1. Please describe the applied minor allele frequency in the manuscript, not only in the review response.

2. A minor allele frequency of 0.01 is likely too small given that the EUR 1000G sample will be about 500. For biallelic variants this implies that there are $0.01^2 * 500 = 0.05$ with 2 copies of the minor allele, and $2 * 0.01 * 0.99 * 500 = 9.9$ people with a single copy of the single allele. This may be too view to get reliable LD estimates. Please consider replicating these analysis with an minor allele cut-off of 0.05 (as a sensitivity analysis). Alternatively, it should be possible to use the available WES data from the UKB which includes thousands of EUR participants.

Version 2:

Reviewer comments:

Reviewer #3

(Remarks to the Author)

No further comments - great work.

We would like to thank the reviewers for their positive, insightful, and constructive comments. In response, we have made several revisions to the manuscript, as outlined below. All changes to the text and figures have been tracked.

Reviewers' comments:

Reviewer #1 (Remarks to the Author):

In the manuscript "LRP5 promotes adipose progenitor cell fitness and adipocyte insulin sensitivity", Loh et al. investigate the role of LRP5 in systemic metabolism, as well as the cellular and molecular mechanisms that underlie the impact of LRP5 on fat distribution. For this purpose, the authors studied gain-of-function and loss-of-function LRP5 variant carriers and performed functional assays in human adipocytes and precursors. The authors provide evidence that LRP5 boosts insulin sensitivity both systemically and within adipocytes. LRP5 plays a vital role in promoting a specific pattern of fat distribution favoring the lower body. It achieves this by preserving the functional traits of adipocytes, partly through activating WNT/ β -catenin signaling and also by maintaining proteostasis via VCP activity, independent of the WNT/ β -catenin pathway. Interestingly, individuals with LRP5 gain-of-function (GoF) mutations were shielded from the typical age-related loss of lower-body fat. The manuscript is well written, and easy to read. The scientific background and rationale for the investigation is clearly addressed. The work is rigorously performed, and interpretation of data is appropriate. This group is a world-leader in this area.

Thank you.

Minor:

1). Can the authors speculate how the depletion of LRP5 causes a decrease in basal glucose uptake in gluteal adipocytes and a decrease in stimulated glucose uptake in abdominal adipocytes?

This is a good point. Following the reviewer's comment, we examined insulin signalling in gluteal LRP5 knockdown adipocytes. Consistent with the stimulated glucose uptake results and in contrast to our findings in abdominal adipocytes, AKT Ser473 phosphorylation in response to insulin stimulation (5 and 10 min, 10 nmol/L insulin) was not impaired in these cells. These data indicate that either LRP5 does not modulate insulin signalling in gluteal adipocytes or, alternatively, that abdominal and gluteal adipocytes have differential insulin sensitivity, possibly with the dose of insulin used obscuring the effect of LRP5 depletion in gluteal cells. This information has now been included in the revised manuscript.

Regarding basal glucose uptake, we examined the expression of *SLC2A1* and *SLC2A3* in abdominal and gluteal DFAT adipocytes. However, we could not detect any differences between LRP5 knockdown abdominal and gluteal adipocytes that could explain the lack of reduction in basal glucose uptake in abdominal cells (Fig. S2). Hence, we revisited our raw data. In our original analysis, we expressed the results as a percentage of vehicle-treated basal cells

for each cell line and analyzed the data using three-way ANOVA (genotype x dox x insulin treatment). This analysis yielded a p-value of 0.18 for the effect of LRP5 knockdown on basal glucose uptake in abdominal adipocytes. However, when we expressed the data as the rate of glucose uptake in pmol/min/mg protein and analyzed the effect of LRP5 knockdown on basal and stimulated glucose uptake separately (two-way ANOVA: genotype x dox), the basal glucose uptake became significant in abdominal adipocytes. We now present these revised analyses in the manuscript and hope that this is acceptable to the reviewer.

2). As a control, it might have been interesting to use purified recombinant Wnts as additional treatments to Chir99021 to evaluate [presumably lack of] functionality of the cells.

We thank the reviewer for this suggestion. Following the reviewer's comment, we investigated the effects of a WNT surrogate-Fc fusion recombinant protein^{1,2} on proliferation and differentiation in abdominal and gluteal control and LRP5 knockdown adipose progenitors (APs). This surrogate consists of the LRP-binding domain of DKK1 linked to a selective, high-affinity binder that interacts with FZD1, FZD2, FZD5, FZD7, and FZD8, and has been shown to recapitulate the WNT3a global transcriptional response in human iPSCs³. All five receptors, as well as *FZD4* and *FZD6* (which are not engaged by the WNT surrogate), are expressed in abdominal and gluteal DFAT APs, with *FZD1* and *FZD7* expression being the highest (our unpublished data).

Using immortalized abdominal and gluteal APs (the parental cell lines used to generate DFAT APs following a cycle of adipocyte differentiation and de-differentiation) constitutively expressing the TOPflash promoter reporter, we found that, similar to CHIR99021, the WNT surrogate was able to activate β -catenin transcriptional responses in these cells (Fig. S7a-b). Subsequently, we tested the effects of 5, 25, and 100 pmol/L concentrations on DFAT AP biology. While the lowest dose had no effects (our unpublished data), treatment with 25 and 100 pmol/L stimulated proliferation in both control and LRP5 knockdown DFAT APs, by presumably engaging LRP6 in the latter since LRP5 protein knockdown was almost complete (Fig. 3b). However, in contrast to CHIR99021, the WNT surrogate further impaired differentiation in LRP5-depleted DFAT APs rather than partially rescuing it. These discrepancies could be due to functional redundancy between LRP5 and LRP6⁴, over-stimulation of WNT/ β -catenin signalling with the WNT surrogate at the doses used, or alternatively, the independence of the effects of CHIR99021 on adipogenesis from WNT/ β -catenin signalling. This information has now been included in the revised results section (Fig. S7) and discussion.

3) For future studies, it would be interesting to isolate cells from adipose tissue of patients and explore directly the functional effects of LRP5 variants.

This is an excellent point. While we no longer have access to these patients, we have immortalized abdominal and gluteal APs from three such individuals. Our future aim is to perform CRISPR editing in these and in existing immortalized APs derived from healthy controls to investigate the direct effects of gain-of-function LRP5 mutations on adipose cell biology.

Reviewer #2 (Remarks to the Author):

Comments to the manuscript COMMSMED-24-0196-T:

The manuscript entitled “LRP5 promotes adipose progenitor cell fitness and adipocyte insulin sensitivity” by Loh NY et al., by studying LRP5 gain- and loss-of-function adipose and metabolic phenotypes, shows that LRP5 promotes lower-body fat distribution and enhances systemic and adipocytes insulin sensitivity. The authors also demonstrate via Mendelian Randomisation analysis and functional studies in LRP5 KD adipocytes, that LRP5’s effects on these traits are mostly cell autonomous and independent of its impact on bones. LRP5 promotes lower-body fat distribution by preserving the functional characteristics of adipose progenitors, partly through WNT/ β -catenin pathway activation, and independently by preserving valosin-containing protein expression and proteostasis. Moreover, both adipocyte and AP LRP5 expression decline with age. Accordingly, gain-of-function LRP5 variant carriers are protected against the age-associated loss of lower-body fat.

Major comments

p.8 line 227. The authors mention “Consistent with these findings, subcutaneous WAT LRP expression...”. The subcutaneous adipose tissue here is the gluteal one? The authors should indicate that in the text. Same for line 234, p.9, the subcutaneous adipose tissue from where is from?

We thank the reviewer for raising this point. By subcutaneous, we mean abdominal and gluteal APs. This information has now been specified in the text to avoid confusion.

Fig.2F. The error bars in this panel are SD or SEM?

The error bars in this panel represent SDs, as now stated in the figure legend.

Fig.2H. The authors should provide quantification of the WB.

This has been done.

Fig.3B. The authors should provide quantification of the WB.

This has been done.

Fig.4E. The authors should provide quantification of the WB.

This has been done.

Fig.5K. The authors should provide quantification of the WB.

This has been done.

Fig.6A, D, G and I. The authors should provide quantification of the WBs.

This has been done.

Reviewer #3 (Remarks to the Author):

The paper nicely combines experimental evidence with evidence from human genetics utilising Mendelian randomisation.

Thank you.

Focus on the latter:

1. The authors describe selecting instruments using a p-value of 5×10^{-8} as well as an F-statistic of 24. This is incorrect, the F-statistic and p-value are proportional to each other and the mentioned thresholds are inconsistent. The aforementioned p-value is equal to an F-statistic of about 29, and the mentioned F-statistic of 24 is equal to a p-value of about 1×10^{-6} . Please fix this and simply mention either the F-statistic or p-value, but not both.

Thank you for pointing this out. Following the reviewer's comment, we have removed the F-statistic values from the Methods section and now mention only the p-values of all the instrumental variables (IVs) used, which were selected based on a threshold of $<5 \times 10^{-8}$.

2. It is not entirely clear but presumably the authors attempted to conduct a cis MR study selecting variants from within and around LRP5? Please clarify and consider referring to the design as a cis-MR study: <https://www.nature.com/articles/s41467-020-16969-0>.

We did not conduct a cis-MR study to explore the possibility that changes in bone mass mediated the effects of LRP5 variants on fat distribution and systemic metabolism. Instead, we felt it was more appropriate to use all GWAS-significant, independent signals associated with heel eBMD in a large UKB GWAS as exposure IVs, rather than GWAS signals solely within the *LRP5* locus. This is particularly relevant as we previously showed, based on cross-sectional epidemiological data, that LRP5 high bone mass (HBM) cases exhibit higher lower-body fat mass compared to non-LRP5 HBM cases^{5,6}, presumably driven by effects of LRP5 on mesenchymal stem cells, which give rise to both osteocytes and adipocytes.

3. Assuming this was a cis-MR why did the authors use a 10Mpb flanking region? Presumably such a large region would include a considerable number of other genes (e.g., about 10 or so) which may give rise to the associations described. Hence the current results may not represent the effects of LRP5. Please either explain why such a large flanking region is used, or consider using a smaller flank (e.g., 100 kbp +- LRP5).

Please see above.

4. Which minor allele frequency was used?

A minor allele frequency threshold of 0.01 was used for instrumental variable selection.

5. Which data were used for the LD calculations and do the authors consider these data sufficiently large to calculate LD for the minor allele frequency chosen (whichever that might be).

We used the 1000 Genomes Phase 3 European reference panel with ~25 million SNPs. It is the standard LD reference panel used in two-sample MR analyses and was also used by the original eBMD GWAS by Morris et al. ⁷.

The IVs for heel eBMD were extracted using the `extract_instruments ()` function in the `TwoSampleMR` package (clumped at LD $r^2 = 0.001$, LD clumping window/genetic distance of 10Mb, using the 1000 Genomes Phase 3 European population reference panel). This information has been included in the Methods section.

6. “An IVW result with $p < 0.05$, coupled with directionally consistent associations in both sensitivity analyses was considered evidence for causal association.” This is a rather liberal selection strategy, would be good to also identify analysis significant in all of the MR analysis, as well as 2 out of 3 (assuming the total number of MR estimator used was 3)?

We now highlight in the text that the associations between BMD and GFAT, and fasting insulin were significant in both the primary univariate IVW MR analyses and in at least one of the two sensitivity analyses, namely MR Egger and Weighted Median MR.

7. Please add units of the exposure when discussing the MR results, as well as in the tables and other display items.

This has been done.

8. Please use scientific notation instead of e-notation throughout.

This has been corrected in Table 3.

9. The authors describe using a two-sample MR design. However going a quick look at the manuscript suggest some of them have included the UKB such a Pulit or Chen (which is the same as used for the exposures). Please consider adding a table with the GWAS studies used and whether they included the UKB or not. If they included the UKB please try to identify the sample overlap.

This is a good point. The GWAS studies used for the MR analyses are listed in Table S2. Where the outcome study also included UKB samples, we have now included a column indicating the percentage of sample overlap.

10. As a complete aside “Statistical analyses were performed using R, Stata, SPSS

and/or GraphPad” using 4 statistic software packages is a bit much – 2 should have been sufficient.

We have removed Stata from the list of statistical software packages used. We also now specify that statistical analyses for the human studies were carried out using SPSS. Statistical analyses for in vitro studies and graph generation were done in GraphPad, while MR and RNA-seq analyses were carried out in R.

References:

- 1 Dang, L. T. *et al.* Receptor subtype discrimination using extensive shape complementary designed interfaces. *Nat Struct Mol Biol* **26**, 407-414, doi:10.1038/s41594-019-0224-z (2019).
- 2 Miao, Y. *et al.* Next-Generation Surrogate Wnts Support Organoid Growth and Deconvolute Frizzled Pleiotropy In Vivo. *Cell Stem Cell* **27**, 840-851 e846, doi:10.1016/j.stem.2020.07.020 (2020).
- 3 <<https://www.thermofisher.com/document-connect/document-connect.html?url=https://assets.thermofisher.com/TFS-Assets/BID/Application-Notes/potent-activation-wnt-pathway-fusion-protein-app-note.pdf>> (Thermofisher).
- 4 Goel, S. *et al.* Both LRP5 and LRP6 receptors are required to respond to physiological Wnt ligands in mammary epithelial cells and fibroblasts. *J Biol Chem* **287**, 16454-16466, doi:10.1074/jbc.M112.362137 (2012).
- 5 Loh, N. Y. *et al.* LRP5 regulates human body fat distribution by modulating adipose progenitor biology in a dose- and depot-specific fashion. *Cell metabolism* **21**, 262-273, doi:10.1016/j.cmet.2015.01.009 (2015).
- 6 Gregson, C. L. *et al.* Mutations in Known Monogenic High Bone Mass Loci Only Explain a Small Proportion of High Bone Mass Cases. *J Bone Miner Res* **31**, 640-649, doi:10.1002/jbmr.2706 (2016).
- 7 Morris, J. A. *et al.* An atlas of genetic influences on osteoporosis in humans and mice. *Nature genetics* **51**, 258-266, doi:10.1038/s41588-018-0302-x (2019).

We would like to thank the reviewers for their positive feedback. In response to the additional points made by reviewer 3, we have revised to the manuscript, as outlined below. All changes to the text and Supplementary table 4 have been tracked.

Reviewers' comments:

Reviewer #1 (Remarks to the Author):

Thank you - this is a terrific paper.

Thank you. We are grateful to the reviewer for the high praise.

Reviewer #2 (Remarks to the Author):

We are satisfied with how the authors addressed our comments. We consider now the manuscript suitable for publication in Communications Medicine.

Thank you.

Reviewer #3 (Remarks to the Author):

Most of my questions have been addressed, although in future it would be beneficial to indicate in the cover letter which changes were made in to manuscript. The following comments remain:

1. Please describe the applied minor allele frequency in the manuscript, not only in the review response.

This information has now been incorporated in the Methods section of the manuscript.

2. A minor allele frequency of 0.01 is likely too small given that the EUR 1000G sample will be about 500. For biallelic variants this implies that there are $0.01^2 * 500 = 0.05$ with 2 copies of the minor allele, and $2 * 0.01 * 0.99 * 500 = 9.9$ people with a single copy of the single allele. This may be too view to get reliable LD estimates. Please consider replicating these analysis with an minor allele cut-off of 0.05 (as a sensitivity analysis). Alternatively, it should be possible to use the available WES data from the UKB which includes thousands of EUR participants.

As suggested by the reviewer, we have repeated all the MR analyses using instrumental variables for heel eBMD with a minor allele cut-off of 0.05 and obtained slightly stronger associations than in the original analysis. This information has now been included in Supplemental table 9 and the Results section.